# Symmetry-Informed Governing Equation Discovery

**Jianke Yang**
UCSD

**Wang Rao**$^*$
Tsinghua University

**Nima Dehmamy**
IBM Research

**Robin Walters**
Northeastern University

**Rose Yu**
UCSD

## Abstract

Despite the advancements in learning governing differential equations from observations of dynamical systems, data-driven methods are often unaware of fundamental physical laws, such as frame invariance. As a result, these algorithms may search an unnecessarily large space and discover less accurate or overly complex equations. In this paper, we propose to leverage symmetry in automated equation discovery to compress the equation search space and improve the accuracy and simplicity of the learned equations. Specifically, we derive equivariance constraints from the time-independent symmetries of ODEs. Depending on the types of symmetries, we develop a pipeline for incorporating symmetry constraints into various equation discovery algorithms, including sparse regression and genetic programming. In experiments across diverse dynamical systems, our approach demonstrates better robustness against noise and recovers governing equations with significantly higher probability than baselines without symmetry. Our codebase is available at https://github.com/Rose-STL-Lab/symmetry-ode-discovery.

## 1 Introduction

Discovering governing equations that can describe and predict observational data from nature is a primary goal of science. Compared to the black-box function approximators common in machine learning, symbolic equations can reveal a deeper understanding of underlying physical processes. In this paper, we consider a first-order dynamical system governed by an autonomous ordinary differential equation (ODE):

$$\dot{\boldsymbol{x}}(t) = \boldsymbol{h}(\boldsymbol{x}(t)) \tag{1}$$

where $t \in T$ denotes time, $\boldsymbol{x}(t) \in X$ is the system state at time $t$, $X \subseteq \mathbb{R}^d$ is the phase space of the ODE, $\dot{\boldsymbol{x}}(t)$ the time derivative, and $\boldsymbol{h} : X \to \mathbb{R}^d$ the vector field describing the evolution dynamics of the system. Our goal is to discover the function $\boldsymbol{h}$ in a symbolic form from the observed trajectories of the system.

Traditionally, discovering governing equations has been a difficult task for human experts due to the complexity of the underlying physical systems and the large search space for equations. Recently, many new algorithms have been proposed for automatically discovering equations from data. When applied to an ODE, these methods first estimate the label, i.e. the time derivative, by numerically differentiating the trajectory. A variety of techniques including sparse function basis regression (Brunton et al., 2016; Champion et al., 2019) and genetic programming (Cranmer et al., 2019) are then used to fit the function between the measurement and the estimated derivative. Another class of methods considers the differential equation's variational form to eliminate the need for derivative estimation and improve the robustness against noise (Messenger & Bortz, 2021; Qian et al., 2022).

An important aspect of equation discovery is the principle of parsimony (Sober, 1981) – *the scientific principle that the most acceptable explanation of a phenomenon is the simplest*. It promotes the scientific goal of identifying a model that offers robust predictive capabilities while maintaining simplicity. Existing methods that fail to consider these principles may end up searching in an

38th Conference on Neural Information Processing Systems (NeurIPS 2024).

unnecessarily large space of equations and discover equations that are overly complex or do not conform to fundamental physical laws.

In this work, we advocate for using the principle of symmetry to guide the equation discovery process. Symmetry has been widely used in machine learning to design equivariant networks (Bronstein et al., 2021; Finzi et al., 2021; Wang et al., 2021), augment training data (Benton et al., 2020), or regularize the model (Akhound-Sadegh et al., 2023; Otto et al., 2023), leading to improved prediction accuracy, sample efficiency, and generalization.

Specifically, we derive the equivariance constraints from the symmetries of ODEs and solve the constraints explicitly to compress the search space of equations. Alternatively, we can also use symmetry as a regularization loss term to guide the equation discovery process and improve its robustness to measurement noise. In practice, a dynamical system may possess symmetries that are unknown or difficult to describe in simple terms. To account for these situations, we also incorporate a method for learning symmetries with nonlinear group actions (Yang et al., 2023a). We then establish a pipeline capable of learning unknown symmetries from data and subsequently discovering governing equations that conform to the discovered symmetry. To summarize, our main contributions include:

- We establish a holistic pipeline to use Lie point symmetries of ODEs to improve the accuracy and robustness of equation discovery algorithms.
- We derive the theoretical criterion for symmetry of time-independent ODEs in terms of equivariance of the associated flow map.
- From this criterion, we solve the linear symmetry constraint explicitly for compressing the equation search space in sparse regression and promoting parsimony in the learned equations.
- For general symmetry, we promote symmetry by a symmetry regularization term when the symmetry constraint cannot be explicitly solved.
- In experiments across many dynamical systems with substantial noise, our symmetry-informed approach achieves higher success rates in recovering the governing equations and better robustness against noise.

## 2 Related Works

### 2.1 Governing Equation Discovery

There are numerous methods for discovering governing equations from data. Genetic programming has been successful in searching exponentially large spaces for combinations of mathematical operations and functions (Gaucel et al., 2014). Recent works have applied genetic programming to distill graph neural networks into symbolic expressions (Cranmer et al., 2019, 2020). Udrescu & Tegmark (2020) introduce physical inductive biases to expedite the search. The main limitation of genetic programming arises from computational complexity due to the combinatorially large search space. Also, genetic programming can suffer from noisy labels (Agapitos et al., 2012), which are common in discovering ODEs due to measurement noise.

Another branch of research involves sparse regression. Originating from SINDy (Brunton et al., 2016), these methods assume that the equation can be written as a linear combination of some predefined functions and optimize the coefficients with sparsity regularization. Subsequently, Champion et al. (2019) combines the sparse regression technique with an autoencoder network to simultaneously discover coordinate systems and governing equations. Weak SINDy (Messenger & Bortz, 2021) uses an alternative optimization objective based on the variational form of the ODE, which eliminates the need for estimating time derivatives and improves the robustness against measurement noise. Other works improve upon SINDy by incorporating physical priors, such as dimensional analysis (Xie et al., 2022; Bakarji et al., 2022), physical structure embedding (Lee et al., 2022), and symmetries (Loiseau & Brunton, 2018; Guan et al., 2021).

### 2.2 Symmetry Prior in Machine Learning

Symmetry plays a crucial role in machine learning. There has been a vast body of work on equivariant neural networks (Cohen & Welling, 2016; Weiler & Cesa, 2019; Finzi et al., 2020, 2021; Bronstein

et al., 2021; Wang et al., 2021). These network architectures enforce various symmetries in different data types. Symmetry can also be exploited through data augmentation (Benton et al., 2020) and canonicalization (Kaba et al., 2023). There are already some works that exploit symmetries for learning in dynamical systems governed by differential equations through data augmentation (Brandstetter et al., 2022), symmetry loss (Huh et al., 2020; Akhound-Sadegh et al., 2023) and contrastive learning (Mialon et al., 2023). However, works using symmetry in recovering underlying equations from data are scarce, and existing examples only consider symmetries specific to a particular system, e.g. reflections and permutations in proper orthogonal decomposition (POD) coefficients (Guan et al., 2021), rotations and translations in space (Ridderbusch et al., 2021; Baddoo et al., 2021), etc. In this work, we provide solutions for dealing with general matrix Lie group symmetries.

While symmetry has proved an important inductive bias, for the ODEs considered in this work, the underlying symmetries are often not known a priori. Some recent works have aimed to automatically discover symmetries from data (Liu & Tegmark, 2022; Yang et al., 2023b,a; Otto et al., 2023; van der Ouderaa et al., 2024). When there is no available prior knowledge about symmetry, We use the adversarial framework in Yang et al. (2023a) to learn unknown symmetries from data and discover the equations subsequently using the learned symmetry.

# 3   Symmetry of ODEs

Our goal is to discover the governing equations of a dynamical system (1) in symbolic form. Our dataset consists of observed trajectories of the system, $\{\boldsymbol{x}_{0:T}^{(i)}\}_{i=1}^N$, where $\boldsymbol{x}_t^{(i)} \in X \subseteq \mathbb{R}^d$ denotes the observation at timestep $t$ in the $i$th trajectory. We assume the observations are collected at a regular time interval $\Delta t$. We omit the trajectory index $i$ and consider one trajectory, $\boldsymbol{x}_{0:T}$ for simplicity. To leverage symmetry for better equation discovery, we first define symmetries in ODEs and provide the theoretical foundation for our methodology. In particular, we show that a time-independent symmetry of the ODE is equivalent to the equivariance of its flow map. This equivariance property can then be conveniently exploited to enforce symmetry constraints in the equation discovery process.

## 3.1   Lie Point Symmetry and Flow Map Equivariance

The Lie point symmetry of an ODE is a transformation that maps one of its solutions to another. The symmetry transformations form the symmetry group of the ODE. In this paper, we focus on symmetry transformations that act solely on the phase space $X \ni \boldsymbol{x}$, without changing the independent variable $t$. We refer to these as *time-independent* symmetries, formally defined as follows.

**Definition 3.1.** A *time-independent* symmetry group of the ODE (1) is a group $G$ acting on $X$ such that whenever $\boldsymbol{x} = \boldsymbol{x}(t)$ is a solution of (1), $\tilde{\boldsymbol{x}} = g \cdot \boldsymbol{x}(t)$ is also a solution of the system.

We refer the readers to Appendix A for background on Lie groups, group actions, and their applications on differential equations. We consider time-independent symmetries in Definition 3.1 and demonstrate how to use these symmetries to derive equivariance constraints on equations.

Given a fixed time $\tau$, the flow map associated with the ODE (1) is denoted $\boldsymbol{f}_\tau : X \to X$, which maps a starting point in a trajectory $\boldsymbol{x}(t_0)$ to an endpoint $\boldsymbol{x}(t_0 + \tau)$ after moving along the vector field $\boldsymbol{h}$ for time $\tau$. Formally, $\boldsymbol{f}_\tau(\boldsymbol{x}_0) = \boldsymbol{x}(\tau)$, where $\boldsymbol{x}(t)$ is the solution of the initial value problem

$$\dot{\boldsymbol{x}}(t) = \boldsymbol{h}(\boldsymbol{x}), \ \boldsymbol{x}(0) = \boldsymbol{x}_0, \ t \in [0, \tau]. \tag{2}$$

**Proposition 3.2.** *Let $G$ be a group that acts on the phase space $X$ of the ODE* (1)*. $G$ is a symmetry group of the ODE* (1) *in terms of Definition 3.1 if and only if for any $\tau \in T$, the flow map $\boldsymbol{f}_\tau$ is equivariant to the $G$-action on $X$.*

As our dataset consists of trajectories measured at discrete timesteps, consider the discretized flow map $\boldsymbol{x}_{t+1} = \boldsymbol{f}(\boldsymbol{x}_t)$, where $\boldsymbol{f} := \boldsymbol{f}_{\Delta t}$ depends on the sampling step size $\Delta t$ of the data. From Proposition 3.2, this function of moving one step forward in the trajectory is $G$-equivariant, i.e.

$$\boldsymbol{x}_{t+1} = \boldsymbol{f}(\boldsymbol{x}_t) \Rightarrow g \cdot \boldsymbol{x}_{t+1} = \boldsymbol{f}(g \cdot \boldsymbol{x}_t). \tag{3}$$

## 3.2 The Infinitesimal Formulae

We can use the equivariance condition (3) to constrain the equation learning problem. Consider the Lie group of symmetry transformations $G$ and the associated Lie algebra $\mathfrak{g}$. We have the following equality constraints.

**Theorem 3.3.** *Let $G$ be a time-independent symmetry group of the ODE* (1). *Let $v \in \mathfrak{g}$ be an element in the Lie algebra $\mathfrak{g}$ of the group $G$. Consider the flow map $\boldsymbol{f}$ defined in* (2) *for a fixed time interval. Denote $J_{\boldsymbol{f}}$ and $J_g$ as the Jacobian of $\boldsymbol{f}(\boldsymbol{x})$ and $g \cdot \boldsymbol{x}$ w.r.t $\boldsymbol{x}$. For all $\boldsymbol{x} \in X$ and $g = \exp(\epsilon v) \in G$, the following equalities hold:*

$$\boldsymbol{f}(g \cdot \boldsymbol{x}) - g \cdot \boldsymbol{f}(\boldsymbol{x}) = 0; \qquad (4) \qquad\qquad J_{\boldsymbol{f}}(\boldsymbol{x})v(\boldsymbol{x}) - v(\boldsymbol{f}(\boldsymbol{x})) = 0; \qquad (5)$$

$$J_g(\boldsymbol{x})\boldsymbol{h}(\boldsymbol{x}) - \boldsymbol{h}(g \cdot \boldsymbol{x}) = 0; \qquad (6) \qquad\qquad J_{\boldsymbol{h}}(\boldsymbol{x})v(\boldsymbol{x}) - J_v(\boldsymbol{x})\boldsymbol{h}(\boldsymbol{x}) = 0. \qquad (7)$$

To understand Theorem 3.3, (4) follows directly from Proposition 3.2, and the other equations are the infinitesimal equivalent of this equivariance condition. For example, the Jacobian-vector product in (5) reveals how much $\boldsymbol{f}$ changes when the input is infinitesimally transformed by $v$, and $v(\boldsymbol{f}(\boldsymbol{x}))$ indicates the amount of change caused by the infinitesimal action on the output of $\boldsymbol{f}$. [2] The difference between these two terms vanishes if the function is equivariant. On the other hand, as both $\boldsymbol{h}$ and $v$ can be viewed as vector fields on $X$, their roles are interchangeable in the equivariance formula. Therefore, we can also express the equivariance in terms of the flow $\boldsymbol{h}$ and a finite group element $g = \exp(\epsilon v)$. Finally, we can also consider the fully infinitesimal formula (7), which indicates that $\boldsymbol{h}$ and $v$ commute as vector fields. The proof of Theorem 3.3 may be found in Appendix B.1.

Theorem 3.3 provides the theoretic basis for our symmetry-based equation discovery methods in Section 4. We can enforce the equality constraints in Theorem 3.3 by solving the constraints explicitly (Section 4.1) or adding them as regularization terms (Section 4.2).

# 4 Symmetry-Informed Governing Equation Discovery

Next, we discuss how to leverage the knowledge of symmetry in equation discovery. Depending on the nature of different symmetries, we develop a holistic pipeline for incorporating symmetries into various equation discovery algorithms, shown in Figure 1. We introduce the components in the following subsections.

## 4.1 `Equiv-c`: Solving the Linear Symmetry Constraint

We start from the case of linear symmetries. Many ODEs exhibit symmetries with linear actions on the state variable $\boldsymbol{x}$. For example, in classical mechanics, the motion of a particle under a central force is invariant under rotations about the center. For such linear symmetries, if the governing equations can be written as linear combinations of basis functions as in SINDy sparse regression (Brunton et al., 2016), the infinitesimal formula (7) becomes a linear constraint. The constraint can be explicitly solved to construct an equivariant model for $\boldsymbol{h}$.

In particular, suppose the governing equations can be written in terms of $\Theta(\boldsymbol{x}) \in \mathbb{R}^p$ as:

$$\boldsymbol{h}(\boldsymbol{x}) = W\Theta(\boldsymbol{x}) \qquad (8)$$

where $W \in \mathbb{R}^{d \times p}$ is the learnable coefficient matrix. In SINDy, the function library $\Theta(\boldsymbol{x})$ is pre-defined. For example, $\Theta$ can be the set of all polynomials up to second order for a 2-dimensional system, i.e. $\Theta(x_1, x_2) = [1, x_1, x_2, x_1^2, x_1 x_2, x_2^2]^T$.

Our goal is to solve for the parameter $W$ that makes (7) hold for any $\boldsymbol{x}$. To this end, we define the symbolic map $M_\Theta : (\mathbb{R}^d \to \mathbb{R}) \to \mathbb{R}^p$, which maps a function to its coordinate in the function space spanned by $\Theta$. Specifically, $M_\Theta(f_j) = \boldsymbol{c}$ for $f_j(\boldsymbol{x}) = \boldsymbol{c}^T\Theta(\boldsymbol{x})$, $\boldsymbol{c} \in \mathbb{R}^p$. For a multivariate function $f : \mathbb{R}^d \to \mathbb{R}^n$, we compute $M_\Theta$ for each of its components and stack them into $M_\Theta(f) \in \mathbb{R}^{n \times p}$. A concrete example is provided in Appendix B.2. We have the following proposition.

**Proposition 4.1.** *Let $G$ be a time-independent symmetry group of the ODE $\dot{\boldsymbol{x}} = W\Theta(\boldsymbol{x})$ with linear actions. Let $v \in \mathfrak{g}$ be an Lie algebra element with action $X$ by $v(\boldsymbol{x}) = L_v\boldsymbol{x}$, $L_v \in \mathbb{R}^{d \times d}$. Then, $L_vW = WM_\Theta(J_\Theta(\cdot)L_v(\cdot))$, where $J_\Theta$ denotes the Jacobian of $\Theta$.*

---

[2]Note that $v(\boldsymbol{f}(\boldsymbol{x}))$ means evaluating the vector field at $\boldsymbol{f}(\boldsymbol{x}) \in X$. It is different from $v(\boldsymbol{f})(\boldsymbol{x})$, which is the Lie derivative of $\boldsymbol{f}$ at the point $\boldsymbol{x}$.

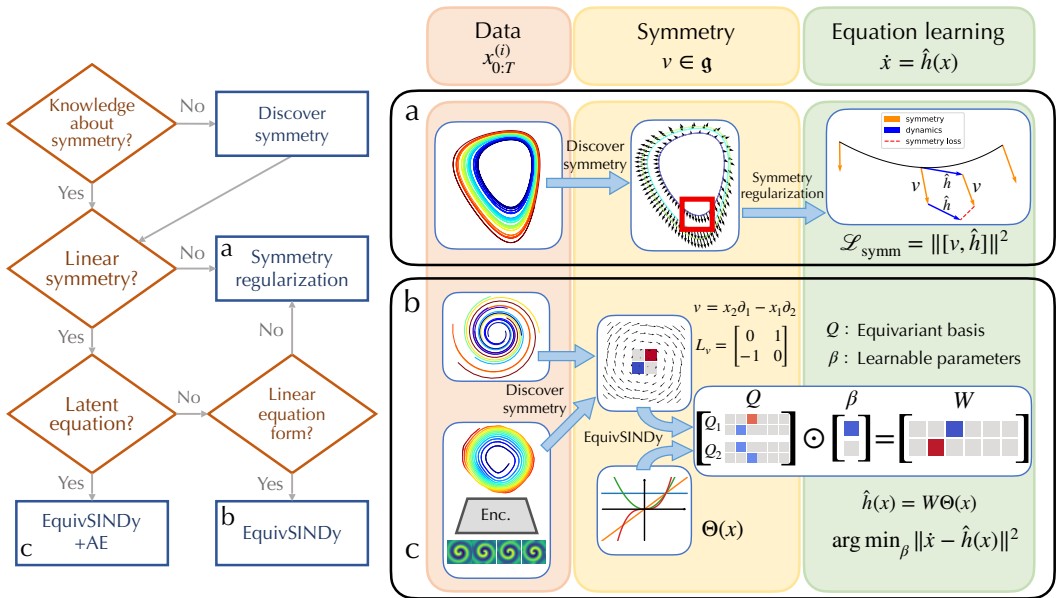

Figure 1: Pipeline for incorporating symmetries into equation discovery via solving linear symmetry constraint (Section 4.1), regularization (Section 4.2) and symmetry discovery (Section 4.3). Given the trajectory data from the dynamical system, we first identify its symmetry based on prior knowledge or symmetry discovery techniques. We then enforce the symmetry by solving a set of constraints when possible and otherwise promote the symmetry through regularization.

*Proof.* Substituting (8) into (7), we obtain $J_v(\boldsymbol{x})W\Theta(\boldsymbol{x}) = W J_\Theta(\boldsymbol{x})v(\boldsymbol{x})$, $\forall \boldsymbol{x} \in X$. As $v$ acts on $X$ linearly, i.e. $v(\boldsymbol{x}) = L_v\boldsymbol{x}$, the constraint is equivalent to $L_v W\Theta(\boldsymbol{x}) = W J_\Theta(\boldsymbol{x})L_v\boldsymbol{x}$.

We then apply $M_\Theta$ to both sides of the equation. Obviously, $M_\Theta(\Theta) = I_p$. Since (7) is true for all $\boldsymbol{x}$, we have $L_v W = W M_\Theta(J_\Theta(\cdot)L_v(\cdot))$. $\qquad\square$

To calculate $M_\Theta$, we need to ensure that $J_\Theta(\boldsymbol{x})L_v\boldsymbol{x}$ is still within the span of the functions in the given $\Theta(\boldsymbol{x})$. This is true when $\Theta(\boldsymbol{x})$ is the set of all polynomials up to a certain degree, which we prove in Appendix B.2.

**Proposition 4.2.** *The components of $J_\Theta(\boldsymbol{x})L_v\boldsymbol{x} \in \mathbb{R}^p$ can be written as linear combinations of the terms in $\Theta(\boldsymbol{x})$ if $\Theta(\boldsymbol{x})$ is the set of all polynomials up to degree $q \in \mathbb{Z}^+$.*

Once we have the infinitesimal action $L_v$ and an appropriate function library $\Theta$, $M_\Theta$ can be computed in a purely symbolic way using `sympy` (Meurer et al., 2017). Denoting $M_{\Theta,L_v} := M_\Theta(J_\Theta(\cdot)L_v(\cdot))$ and vectorizing the matrix $W$, we can rewrite the equation in Proposition 4.1 as the following linear constraint on $W$:

$$(-M_{\Theta,L_v}^T \bar{\oplus} L_v)\text{vec}(W) = 0 \tag{9}$$

where $\bar{\oplus}$ is the Kronecker sum: $A \bar{\oplus} B = A \otimes I + I \otimes B$. We concatenate the constraints for all representations $L_{v_i}$ of the Lie algebra basis $\{v_i\}_{i=1}^c$ into a single matrix $C$ and solve the constraint using the singular value decomposition:

$$
C\,\text{vec}(W) = \begin{bmatrix} -M_{\Theta,L_{v_1}}^T \bar{\oplus} L_{v_1} \\ \dots \\ -M_{\Theta,L_{v_c}}^T \bar{\oplus} L_{v_c} \end{bmatrix} \text{vec}(W)
$$

$$
= U \begin{bmatrix} \Sigma & 0 \\ 0 & 0 \end{bmatrix} \begin{bmatrix} P^T \\ Q^T \end{bmatrix} \text{vec}(W) = 0. \tag{10}
$$

The coefficient matrix resides in the $r$-dimensional nullspace of $C$ and can be parametrized as $\text{vec}(W) = Q\beta$, $\beta \in \mathbb{R}^r$. This equivariant parametrization significantly reduces the number of free parameters from $d \times p$ to $r$, leading to parsimonious equations and easier training in a compressed parameter space. We name this approach as `EquivSINDy-c`, where c stands for **c**onstraint.

The above procedure uses linear symmetries to inform sparse regression. However, when the data is high-dimensional, such as videos, linear symmetries can become too restrictive. For these scenarios, SINDy autoencoder (Champion et al., 2019) maps the data to a latent space to discover the coordinate system and the governing equation under that system. We show in the experiment section that our model can also be used in this case by enforcing the latent equation to be equivariant. With the inductive bias of symmetry, the resulting coordinate and equation can achieve better long-term prediction accuracy.

## 4.2 `Equiv-r`: Regularization for General Symmetry

The above procedure, `EquivSINDy-c`, applies to linear symmetry with a proper choice of function library $\Theta$. Solving the equality constraint in Proposition 4.1 for arbitrary group action is generally more challenging. Technically, when the infinitesimal action $v$ is expressed in a closed form, we can still apply $M_\Theta$ to both sides of the equation and obtain $M_\Theta(v(W\Theta(\cdot))) = WM_\Theta(J_\Theta(\cdot)v(\cdot))$. However, the function library $\Theta$ needs to be carefully chosen based on the specific action $v$ to ensure that $M_\Theta$ is evaluated on functions within the span of $\Theta(\boldsymbol{x})$. Moreover, as we will discuss in Section 4.3, we may rely on symmetry discovery methods to learn unknown symmetries parametrized by neural networks. Without a closed-form symmetry, calculating $M_\Theta$ is intractable.

We propose an alternative approach that is universally applicable to any symmetry and equation discovery algorithm. We use the formula from Theorem 3.3 as a regularization term to promote symmetry in the learned equation. This symmetry loss is added to the equation loss from the base equation discovery algorithm, e.g. $L_2$ error between the estimated time derivative and the prediction from the learned equation.

We consider the following relative loss based on infinitesimal group action $v$ and a finite-time flow map $\boldsymbol{f}_\tau$ of the equation as in (5):

$$\mathcal{L}_{\text{symm}} = \mathbb{E}_{\boldsymbol{x}} \left[ \sum_{v \in B(\mathfrak{g})} \frac{\|J_{\boldsymbol{f}_\tau}(\boldsymbol{x})v(\boldsymbol{x}) - v(\boldsymbol{f}_\tau(\boldsymbol{x}))\|^2}{\|J_{\boldsymbol{f}_\tau}(\boldsymbol{x})v(\boldsymbol{x})\|^2} \right] \tag{11}$$

where $B(\mathfrak{g})$ denotes the basis of the Lie algebra, $\boldsymbol{f}_\tau$ is obtained by solving the initial value problem (2), and $J_{\boldsymbol{f}_\tau}$ denotes the Jacobian of $\boldsymbol{f}_\tau$.

We use a relative loss because the scale of both terms in the numerator and their difference is subject to the specific $\boldsymbol{f}_\tau$. In the extreme case when $\boldsymbol{h} = \boldsymbol{0}$, both terms are zero because $\boldsymbol{f}_\tau(\boldsymbol{x})$ is constant. As we are optimizing $\boldsymbol{h}$ under this objective, we compute the relative error to eliminate the influence of the level of variations in the function itself on the symmetry error.

We can also introduce other forms of symmetry regularizations based on the other equations in Theorem 3.3. Empirically, these regularization terms perform similarly, but their implementations have subtle differences. For instance, some loss terms require integrating the learned equations to get the flow map $\boldsymbol{f}_\tau$ and thus incur additional computational cost; some involve computing higher-order derivatives for infinitesimal symmetries, which amplifies the numerical error when the symmetry is not exactly accurate. We list these different options of symmetry regularization and compare them in more detail in Appendix C.2.

The use of symmetry regularization encourages equation discovery models to achieve lower symmetry error, instead of explicitly constraining the parameter space as in Section 4.1. However, it is a general approach that can be applied to various equation discovery algorithms, e.g. sparse regression and genetic programming. Similar to the works that utilize the variational form of ODEs (Messenger & Bortz, 2021; Qian et al., 2022), our symmetry regularization does not require estimating the time derivative, so it is more robust to noise. We name this approach as `EquivSINDy-r` (or `EquivGP-r` for genetic programming-based discovery algorithms), `r` standing for **r**egularization.

## 4.3 Equation Discovery with Unknown Symmetry

The knowledge of ODE symmetries is often not accessible when we do not know the equations. In this case, we can use symmetry discovery methods (Dehmamy et al., 2021; Liu & Tegmark, 2022; Yang et al., 2023b,a; Otto et al., 2023) to first learn the symmetry from data and then use the discovered symmetries to improve equation discovery.

In our experiments, we use LaLiGAN (Yang et al., 2023a) to learn nonlinear actions of a Lie group $G \leq \mathrm{GL}(k; \mathbb{R})$ as $\exp(\epsilon v)x = (\psi \circ \exp(\epsilon L_v) \circ \phi)(x)$, where the networks $\psi$ and $\phi$ define an autoencoder and $L_v \in \mathbb{R}^{k \times k}$ is the matrix representation of $v \in \mathfrak{g}$. Our dataset consists of ODE trajectories sampled at a fixed rate $\Delta t$, each given by $\boldsymbol{x}_{0:T}$. We can learn the equivariance of $\boldsymbol{f} : \boldsymbol{x}_t \mapsto \boldsymbol{x}_{t+1}$ by feeding the input-output pairs $(\boldsymbol{x}_i, \boldsymbol{x}_{i+1})$ into LaLiGAN. We extract the infinitesimal action as

$$v(\boldsymbol{x}) = \frac{d}{d\epsilon}(\psi \circ \exp(\epsilon L_v) \circ \phi)(\boldsymbol{x})\Big|_{\epsilon=0} = J_\psi(\boldsymbol{z})L_v\boldsymbol{z} \tag{12}$$

where $\boldsymbol{z} = \phi(\boldsymbol{x})$ is the latent mapping via encoder network, and the Jacobian-vector product can be obtained by automatic differentiation. This infinitesimal action is then used to compute the regularization (11).

Alternatively, we can also discover equations in the latent space of LaLiGAN by solving the equivariance constraints in Section 4.1, as the latent dynamics $\boldsymbol{z}_t \mapsto \boldsymbol{z}_{t+1}$ has linear symmetry $L_v$. Discovering governing equations in the latent space can be helpful when the dataset consists of high-dimensional observations instead of low-dimensional state variables.

We should note that our method has a completely different goal from LaLiGAN. Our method aims to discover equations using symmetry as an inductive bias. LaLiGAN aims to discover unknown symmetry. Only when we do not know the symmetry a priori, we use LaLiGAN as a tool to discover the symmetry first and use the discovered symmetry to regularize equation learning.

## 5 Experiments

The experiment section is organized as follows. In Section 5.1, we will demonstrate how to solve the linear symmetry constraints (Section 4.1) on some synthetic equations. In Section 5.2, we will apply the same equivariant model to discover equations in a symmetric latent space for some high-dimensional observations. In Section 5.3, we consider two well-studied benchmark problems for dynamical system inference under significant noise levels, where we use the regularization technique (Section 4.2) based on learned symmetries.

**Data generation.** For each ODE system, we sample $N$ random initial conditions $\boldsymbol{x}_0$ from a uniform distribution on a specified subset of $X \subseteq \mathbb{R}^d$. Starting from each initial condition, we integrate the ODE using the 4th-order Runge-Kutta (RK4) method and sample with a regular step size $\Delta t$ to get the discrete trajectory $\boldsymbol{x}_{0:T}$. We use different $N$, $T$ and $\Delta t$ for each system, as specified in Appendix D.

Then, unless otherwise specified, we add white noise to each dimension of the state $\boldsymbol{x}$ at a noise level $\sigma_R$. The scale of the noise depends on the variance in the data within each state dimension $x_i$: $\sigma_i = \sigma_R \cdot \mathrm{std}(x_i)$. We will report the noise level $\sigma_R$ for each system. After adding the noise, we apply a Gaussian process-based smoothing procedure on the noisy data, similar to Qian et al. (2022).

**Evaluation metrics.** We run each algorithm multiple times and report the *success probability* of discovering the correct function form. Specifically, assume the true equation is expanded as $\sum \theta_{f_i} f_i(\boldsymbol{x})$, where $\theta_{f_i}$ is a nonzero constant parameter and $f_i$ is a function of $\boldsymbol{x}$. Also, we expand the discovered equation as $\sum \hat{\theta}_{f_i} \hat{f}_i(\boldsymbol{x})$, where $\hat{\theta}_{f_i} \neq 0$. The discovery is considered successful if all the terms match, i.e. $\{f_i\} = \{\hat{f}_i\}$. The probability is computed as the proportion of successful runs. As each dynamical system consists of multiple equations (one for each dimension), we evaluate the success probability of discovering each individual equation (e.g. "Eq. 1", "Eq. 2" columns in Table 1) and the joint probability of successfully discovering all the equations (e.g. "All" column in Table 1). We argue that this is the most important metric for evaluating the performance of an equation discovery algorithm, since an accurate equation form reveals the key variables and their interactions, and thus the underlying structure and relationships governing the system.

The following metrics are also considered:

- *RMSE* for parameter estimation $\sqrt{\sum \|\boldsymbol{\theta} - \hat{\boldsymbol{\theta}}^{(k)}\|^2/K}$, where $K$ is the number of runs, $\boldsymbol{\theta} = (\theta_{f_1}, \theta_{f_2}, ...)$ is the constant parameters in true equation terms as defined above, and

$\hat{\boldsymbol{\theta}}^{(k)}$ the estimated parameters of the corresponding terms from the $k$th run. If $f_i$ is not in the learned equation, $\hat{\theta}_{f_i} = 0$. We evaluate this metric on (1) all runs and (2) successful runs, referred to as *RMSE (all)* and *RMSE (successful)*. Also, for each system, this metric is computed for each individual equation and all equations.

- *Long-term prediction error.* We integrate the learned ODE from an initial condition sampled from the test set to predict the future trajectory. We evaluate the error between the predicted and the true trajectory at chosen timesteps.

**Algorithms and Baselines.** For baseline comparisons, we include SINDy sparse regression (Brunton et al., 2016) and Genetic Programming (GP) with `PySR` package (Cranmer, 2023), and their respective variants using the variational formulation, i.e. Weak SINDy (WSINDy) (Messenger & Bortz, 2021) and D-CODE (Qian et al., 2022). Based on different experiment setups and the guidelines in Figure 1, we apply our methods referred to as `EquivSINDy` / `EquivGP`, where the suffix `c` stands for solving the linear symmetry constraint, and `r` stands for using the symmetry regularization.

## 5.1 Equivariant SINDy for Linear Symmetries

In this section, we consider two dynamical systems which possess some linear symmetries. These symmetries are commonly observed across many systems and their presence can be easily detected. Therefore, we assume we know these symmetries and focus on symmetry-informed equation discovery by solving the symmetry constraints.

Figure 2: Solutions for equivariant constraints of (13, 14). In (13), the 2D parameter space is spanned by $Q_1$ and $Q_2$. In (14), the 3D parameter space is spanned by $Q_{1,2,3}$, each marked with a different color.

**Damped Oscillator.** This is a two-dimensional system with rotation symmetry, characterized by the infinitesimal generator $v = x_2 \partial_1 - x_1 \partial_2$ (i.e. $\begin{bmatrix} 0 & 1 \\ -1 & 0 \end{bmatrix}$ in matrix terms). It is governed by (13).

$$\begin{cases} \dot{x}_1 = -0.1x_1 - x_2 \\ \dot{x}_2 = x_1 - 0.1x_2 \end{cases} \quad (13) \qquad \begin{cases} \dot{x}_1 = -0.3x_1 + 0.1x_2^2 \\ \dot{x}_2 = x_2 \end{cases} \quad (14)$$

**Growth.** This system exhibits a scaling symmetry under $(x_1, x_2) \mapsto (a^2 x_1, a x_2)$. The corresponding infinitesimal generator is $v = 2x_1 \partial_1 + x_2 \partial_2$. It is governed by (14).

Figure 2 visualizes the parameter spaces for these two systems under the equivariance constraint. The original parameter space is $W \in \mathbb{R}^{2 \times 6}$ as we build the function library $\Theta$ with up to second-order polynomials. For (13), the equivariance constraint reduces the parameter space to two dimensions. The equivariant basis $Q_1 = Qe_1$ and $Q_2 = Qe_2$ are visualized. For (14), the parameter space is reduced to three dimensions. Each equivariant basis component only affects one equation term, visualized in three distinct colors. We can observe that the equivariant models have much fewer free parameters than non-equivariant ones.

Table 1: Success probability of equation discovery on the damped oscillator (13) at noise level $\sigma_R = 20\%$ and the growth system (14) at $\sigma_R = 5\%$, computed from 100 runs for each algorithm. See Appendix C.1 for full results.

| System | Method | Success prob. | | |
|---|---|---|---|---|
| | | Eq. 1 | Eq. 2 | All |
| Oscillator (13) | GP | 0.00 | 0.70 | 0.00 |
| | D-CODE | 0.00 | 0.00 | 0.00 |
| | SINDy | 0.35 | 0.38 | 0.15 |
| | WSINDy | 0.06 | 0.07 | 0.00 |
| | EquivSINDy-c | **0.93** | **0.97** | **0.90** |
| Growth (14) | GP | 0.00 | **1.00** | 0.00 |
| | D-CODE | 0.00 | 0.65 | 0.00 |
| | SINDy | 0.26 | 0.13 | 0.03 |
| | WSINDy | 0.00 | 0.00 | 0.00 |
| | EquivSINDy-c | **1.00** | **1.00** | **1.00** |

Table 1 displays the success probability of different methods. With the reduced weight space, our equivariant model almost always recovers the correct equation forms. Full results, including the parameter estimation error and the prediction error with discovered equations, as well as the experiment on a higher-dimensional system, are deferred to Appendix C.1. Overall, these results fully demonstrate the advantage of applying symmetry to compress the parameter space.

## 5.2 Equivariant SINDy in Latent Space

We also show an example of discovering equations in the latent space with an equivariant model. We consider a $\lambda - \omega$ reaction-diffusion system visualized in Figure 3. The system is sampled at a $100 \times 100$ spatial grid, yielding observations in $\mathbb{R}^{10000}$. See Appendix D.1 for more details.

We apply LaLiGAN (Yang et al., 2023a) to discover a latent space $\mathbb{R}^2$ where the dynamics $z_t \mapsto z_{t+1}$ is equivariant to a linear group action. Meanwhile, we use equivariant SINDy with the learned symmetry to discover the equation for the latent dynamics. We compare our approach with SINDy Autoencoder (Champion et al., 2019) without any symmetry and LaLiGAN + SINDy, i.e. learning the equations in the LaLiGAN latent space but without equivariance constraint.

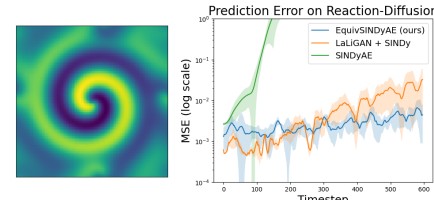

Figure 3: Reaction-diffusion system and the prediction error in the high-dimensional input space using equations from different methods. The means and standard deviations (shaded area) of errors over 3 random runs are plotted.

Figure 3 shows the prediction error on this system using the latent equations. We find that the discovery result of SINDy Autoencoder is unstable, and the simulation can quickly diverge. Meanwhile, both approaches with symmetry discover equations that accurately simulate the dynamics in the input space. Interestingly, even with similar latent spaces discovered by LaLiGAN, our approach that enforces the equivariance constraint can further reduce prediction error compared to non-constrained regression.

## 5.3 Symmetry Regularization

Table 2: Equation discovery statistics on the Lotka-Volterra system (15) at noise level $\sigma_R = 99\%$ and the glycolytic oscillator (16) at noise level $\sigma_R = 20\%$. The RMSE is scaled by $\times 10^{-1}$ for (15) and $\times 10^{-2}$ for (16). The success probability is computed from 50 runs for sparse regression-based algorithms and 10 runs for genetic programming-based algorithms. The success probabilities of recovering individual equations (Eq. 1 & Eq. 2) and simultaneously recovering both equations (All) are reported. The RMSE (all) refers to the parameter estimation error over all runs. The RMSE (successful) refers to the parameter estimation error over successful runs, which is missing for algorithms with zero success probability.

| System | Method | Success prob. | | | RMSE (successful) | | | RMSE (all) | |
|---|---|---|---|---|---|---|---|---|---|
| | | Eq. 1 | Eq. 2 | All | Eq. 1 | Eq. 2 | All | Eq. 1 | Eq. 2 |
| L-V (15) | SINDy | 0.40 | **0.64** | 0.24 | 1.01 (0.26) | 0.56 (0.21) | 0.79 (0.16) | 4.01 (2.50) | **3.24** (3.67) |
| | WSINDy | 0.18 | 0.22 | 0.06 | **0.59** (0.38) | **0.32** (0.23) | **0.26** (0.16) | 16.13 (13.28) | 18.66 (18.81) |
| | EquivSINDy-r | **0.54** | 0.58 | **0.36** | 1.00 (0.21) | 0.45 (0.20) | 0.79 (0.15) | **3.16** (2.46) | 3.83 (4.01) |
| | GP | **1.0** | 0.0 | 0.0 | **2.44** (0.89) | N/A | N/A | **2.44** (0.89) | 13.20 (3.20) |
| | D-CODE | 0.0 | 0.0 | 0.0 | N/A | N/A | N/A | 10.38 (0.11) | 9.08 (1.88) |
| | EquivGP-r | **1.0** | **0.8** | **0.8** | **2.43** (1.39) | **0.51** (0.98) | **1.58** (1.62) | **2.43** (1.39) | **1.76** (0.31) |
| Glycolytic Oscillator (16) | SINDy | 0.30 | 0.56 | 0.14 | 0.87 (0.15) | 0.32 (0.13) | 0.67 (0.10) | 15.86 (17.52) | 12.71 (18.32) |
| | WSINDy | 0.06 | 0.14 | 0.04 | **0.11** (0.10) | 0.59 (0.23) | **0.34** (0.04) | 2.3e3 (2.9e3) | 2.1e3 (2.7e3) |
| | EquivSINDy-r | **0.40** | **0.70** | **0.28** | 0.92 (0.22) | **0.30** (0.13) | 0.71 (0.16) | **9.97** (8.07) | **7.29** (12.72) |
| | GP | 0.0 | 0.0 | 0.0 | N/A | N/A | N/A | **7.57** (5.46) | 14.88 (23.50) |
| | D-CODE | 0.0 | 0.4 | 0.0 | N/A | 0.27 (0.11) | N/A | 14.37 (15.66) | 8.90 (16.84) |
| | EquivGP-r | **0.1** | **0.6** | **0.1** | **0.67** (N/A) | **0.22** (0.38) | **0.40** (N/A) | 8.59 (12.10) | **3.43** (6.69) |

We demonstrate the use of symmetry regularization in the following ODE systems. These systems do not possess any linear symmetries for solving the constraint explicitly.

**Lotka-Volterra System** describes the interaction between a predator and a prey population. We consider its canonical form as in Yang et al. (2023a), where the state is given by the logarithm population densities of the prey and the predator.

$$\begin{cases} \dot{x}_1 = \dfrac{2}{3} - \dfrac{4}{3}e^{x_2} \\ \dot{x}_2 = -1 + e^{x_1} \end{cases} \tag{15}$$

**Glycolytic Oscillator** (Sel'Kov, 1968) is a biochemical system of two ODEs with complex interactions in cubic terms. We use the same constants in the equations as in Qian et al. (2022).

$$\begin{cases} \dot{x}_1 = 0.75 - 0.1x_1 - x_1x_2^2 \\ \dot{x}_2 = 0.1x_1 - x_2 + x_1x_2^2 \end{cases} \tag{16}$$

Since we assume no knowledge of the symmetries here, we first run LaLiGAN (Yang et al., 2023a) to discover symmetry from data for each system. We then extract the group action and the induced infinitesimal action from LaLiGAN and use them to compute symmetry regularization. Table 2 shows the results of different discovery algorithms with and without symmetry. Comparison is made within each class of methods, i.e. sparse regression and genetic programming. For genetic programming, we use an alternate form of symmetry regularization (37), which will be discussed in Appendix C.2.

Informed by symmetries through regularization terms, equation discovery algorithms have a much higher success probability of discovering the correct equation forms. Also, models with symmetry regularization generally have the lowest parameter estimation errors averaging over all random experiments.

In Appendix C.2, we provide additional results to show that the equations discovered by our method also achieve lower prediction error. We also compare the different regularization options based on Theorem 3.3.

## 6   Conclusion

We propose to incorporate symmetries into equation discovery algorithms. Depending on whether the action of symmetry is linear, we develop a pipeline (Figure 1) with various techniques, e.g. solving the equivariance constraint and using different forms of symmetry regularization, to perform symmetry-informed equation discovery. Experimental results show that our method can reduce the search space of equations and recover true equations from noisy data.

Our methodology applies to time-independent symmetries for autonomous ODEs. These symmetries have relatively simple forms and can be either obtained as prior knowledge or learned by existing algorithms. In theory, our method can be generalized to other settings, such as time-dependent symmetries and non-autonomous ODEs. We provide a more general problem definition and discuss these potential generalizations in Appendix A.2. However, in these problems, it is less likely that we know the symmetries before discovering the equations. Extending our symmetry-informed methodology to these scenarios would require further breakthroughs in learning symmetries for general differential equation systems, which could be an interesting future research direction.

## Acknowledgement

This work was supported in part by the U.S. Army Research Office under Army-ECASE award W911NF-23-1-0231, the U.S. Department Of Energy, Office of Science, IARPA HAYSTAC Program, CDC-RFA-FT-23-0069, DARPA AIE FoundSci, DARPA YFA, NSF Grants #2205093, #2146343, and #2134274.

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

# A  Preliminary on Lie Point Symmetry of Differential Equations

We use the theory of Lie groups to describe the symmetry transformations of ODE solutions. This section provides a brief overview of some fundamental concepts of Lie group and its application to differential equations.

## A.1  Lie Groups

**Lie groups.**  A Lie group is a smooth manifold with a compatible group structure. We use Lie groups to describe continuous transformations. For example, all $n \times n$ invertible matrices with real entries form the general linear group $\mathrm{GL}(n; \mathbb{R})$ with matrix multiplication as the group operation.

The Lie algebra $\mathfrak{g} := T_e G$ is defined as the tangent space at the identity element $e$ of the group $G$. For example, The Lie algebra of general linear group $\mathrm{GL}(n, \mathbb{R})$ consists of all real-valued matrices of size $n \times n$. We can use a vector space basis $\{v_i \in \mathfrak{g}\}$ to describe any Lie algebra element as $v = \sum \epsilon_i v_i$, where $\epsilon_i \in \mathbb{R}$. Lie algebra can be thought of as the space of infinitesimal transformations of the group. We can map the Lie algebra elements to group elements (in the identity component) via the exponential map, $\exp \colon \mathfrak{g} \to G$. For matrix Lie groups, matrix exponential is such a map.

**Group actions.**  A group action defines how abstract group elements transform other objects in a space $V$. The group action is a function $\Gamma \colon G \times V \to V$ which maps the identity $e$ to identity transformation, i.e. $\Gamma(e, v) = v$, and is compatible with group element composition, i.e. $\Gamma(g_1, \Gamma(g_2, v)) = \Gamma(g_1 g_2, v)$. We can also think of the group action as mapping a group element to a function $V \to V$ and write $\Gamma_g := \Gamma(g, \cdot)$. The group action $\Gamma$ induces an infinitesimal action $\gamma$ of its Lie algebra given by $\gamma_v(x) = \frac{d}{d\epsilon}(\Gamma_{\exp(\epsilon v)}(x))|_{\epsilon=0}$. For a smooth manifold $V$, the infinitesimal action can be interpreted as a vector field on $V$ that generates the flow coinciding with the group action of $\exp(\epsilon v)$. Given the coordinate $(x_1, ..., x_d)$, the vector field can be written as $\gamma_v = \sum_i v^i \partial_i$, where $\partial_i \equiv \frac{\partial}{\partial x_i}$ and $v^i \colon V \to \mathbb{R}$ are the components of $\gamma_v$.

A group action is called a representation if $V$ is a vector space and $\Gamma_g$ is a linear function on $V$. When $V = \mathbb{R}^n$, the representation $\Gamma_g \in \mathrm{GL}(n; \mathbb{R})$ is an invertible $n \times n$ matrix that transforms $v \in V$ by matrix multiplication. Correspondingly, the induced Lie algebra representation $\gamma_v \in \mathfrak{gl}(n, \mathbb{R})$ is also an $n \times n$ matrix. In the following discussion, we will refer to the symmetries with linear actions as linear symmetries and those with nonlinear actions as nonlinear symmetries.

**Notations.**  In this paper, we mostly consider the group action and the corresponding infinitesimal action on the phase space of the ODE, $X$. For simplicity, we use $g \in G$ to refer to both the group element and the group action $\Gamma_g$ and write $g \cdot \boldsymbol{x} := \Gamma_g(\boldsymbol{x})$. Similarly, we use $v \in \mathfrak{g}$ to refer to both the Lie algebra element and its infinitesimal action and write $v(\boldsymbol{x}) := \gamma_v(\boldsymbol{x})$.

## A.2  Lie Point Symmetry of ODEs

### A.2.1  ODE

In this work, we are mainly concerned with the autonomous ODEs in the form (1). However, to introduce the general notion of Lie point symmetry of ODEs, it is helpful to consider a more generalized setup as follows:

$$\frac{d\boldsymbol{x}}{dt} = \boldsymbol{h}(t, \boldsymbol{x}) \tag{17}$$

where $t \in T \subseteq \mathbb{R}$ denotes time, i.e. the only independent variable in ODEs, and $\boldsymbol{x} \in X \subseteq \mathbb{R}^d$ denotes the dependent variables.

To formalize the concept of symmetries, we take a geometric perspective on the differential equations (17). The independent and the dependent variables form a total space $E = T \times X$. We define the first-order jet space of the total space as $\mathcal{J} = T \times X \times X_1$. Its coordinates, $(t, x_1, ..., x_n, x'_1, ..., x'_n) \in \mathcal{J}$, represent the independent and the dependent variables as well as the time derivatives of the dependent variables. This is also known as the first-order **prolongation**.

Then, we can represent the ODE (17) through the map $\Delta : \mathcal{J} \to \mathbb{R}^n$ with components $\Delta_i = x'_i - h_i(t, \boldsymbol{x})$. A function $\boldsymbol{x}(t)$ is a solution of (17) if its prolongation satisfies

$$\Delta(t, \boldsymbol{x}, \boldsymbol{x}') = 0, \ \forall t \tag{18}$$

### A.2.2 Symmetry

A symmetry of the system $\Delta$ transforms one of its solutions to another. More specifically, we need to define the group action on a point in the total space $E = T \times X$ and its induced action on a function $T \to X$.

We express the action of group element $g$ in the total space as

$$g \cdot (t, \boldsymbol{x}) = (\hat{t}(t, \boldsymbol{x}), \hat{\boldsymbol{x}}(t, \boldsymbol{x})), \tag{19}$$

where $\hat{t}$ and $\hat{\boldsymbol{x}}$ are functions over $T \times X$.

To define the induced group action on the function $f : T \to X$, we first identify $f$ with its graph

$$\gamma_f = \{(t, f(t)) : t \in \Omega_T\} \subset T \times X \tag{20}$$

where $\Omega_T \subset T$ is the time domain. The graph of the function can be transformed by $g$ as

$$g\gamma_f = \{(\tilde{t}, \tilde{\boldsymbol{x}}) = g \cdot (t, \boldsymbol{x}) : (t, \boldsymbol{x}) \in \gamma_f\} \tag{21}$$

The set $g\gamma_f$ may not be the graph of another single-valued function. However, by choosing a suitable domain $\Omega_T$, we ensure that the group elements near the identity transform the original graph into a graph of some other single-valued function $\tilde{\boldsymbol{x}} = \tilde{f}(\tilde{t})$ with the transformed graph $\gamma_{\tilde{f}} = g \cdot \gamma_f$. In other words, the transformed function $\tilde{f} = g \cdot f$ is defined by the transformation of the function graph.

**Definition A.1** (Def 2.23, Olver (1993)). A symmetry group of $\Delta$ (18) is a local group of transformations $G$ acting on an open subset of the total space $T \times X$ that whenever $\boldsymbol{x} = f(t)$ is a solution of $\Delta$, and whenever $g \cdot f$ is defined for $g \in G$, then $\boldsymbol{x} = (g \cdot f)(t)$ is also a solution of the system.

### A.3 The Infinitesimal Criterion

Let $v$ be a vector field on the total space $T \times X$ with corresponding one-parameter subgroup $\exp(\epsilon v)$. The vector field describes the infinitesimal transformations to the independent and the dependent variables. It can be written as

$$v(t, \boldsymbol{x}) = \xi(t, \boldsymbol{x})\partial_t + \sum_i \phi^i(t, \boldsymbol{x})\partial_i, \ \partial_i \equiv \frac{\partial}{\partial x_i}. \tag{22}$$

The vector field also has its induced infinitesimal action on the jet space $T \times X \times X_1$ described by the **prolonged vector field**:

$$v^{(1)} = v + \sum_i \phi_i^{(1)}(t, \boldsymbol{x}, \boldsymbol{x}')\partial_{x'_i} \tag{23}$$

where $\phi_i^{(1)}$ can be computed by the *prolongation formula* (Theorem 2.36, Olver (1993)) as

$$\phi_i^{(1)} = D_t[\phi_i - \xi x'_i] + \xi x''_i \tag{24}$$

$$= \frac{\partial \phi_i}{\partial t} + \sum_j x'_j \frac{\partial \phi_i}{\partial x_j} - x'_i \left( \frac{\partial \xi}{\partial t} + \sum_j x'_j \frac{\partial \xi}{\partial x_j} \right) \tag{25}$$

We denote $\phi(t, \boldsymbol{x}) \coloneqq [\phi_1(t, \boldsymbol{x}), ..., \phi_d(t, \boldsymbol{x})]^T$ as the vector field components in $X$.

As a special case, for the time-independent symmetries we considered previously, $\xi = 0$ and $\phi(t, \boldsymbol{x}) = \phi(\boldsymbol{x})$, so we have

$$\phi_i^{(1)} = \sum_j \frac{\partial \phi_i}{\partial x_j} x'_j \tag{26}$$

Another special case is time translation. When $\xi = 1$ and $\phi = 0$, we have $\phi_i^{(1)} = 0$.

The following infinitesimal criterion gives the symmetry condition in terms of the vector field (and its prolongation).

**Theorem A.2** (The infinitesimal criterion. Thm 2.31, Olver (1993)). *Let $\Delta$ be a system of ODEs of maximal rank defined over $M \subset T \times X$, $X \subseteq \mathbb{R}^d$. A local group of transformations $G$ acting on $M$ is a symmetry group of the system $\Delta$ iff [3] for every infinitesimal genrator $v$ of $G$, $v^{(1)}[\Delta_i] = 0$, $i = 1, ..., d$, whenever $\Delta = 0$.*

Once we have the symmetry represented by the vector field $v$, we can derive constraints on the equation form from the above theorem.

Generally, when the vector field takes the general form in (22), Theorem A.2 is equivalent to

$$J_\phi(t) + J_\phi(\boldsymbol{x})\boldsymbol{h}(\boldsymbol{x}) - (\xi_t + (\nabla_{\boldsymbol{x}}\xi)^T \boldsymbol{h}(\boldsymbol{x}))\boldsymbol{h}(\boldsymbol{x}) = J_{\boldsymbol{h}}(t, \boldsymbol{x}) \begin{bmatrix} \xi(t, \boldsymbol{x}) \\ \phi(t, \boldsymbol{x}) \end{bmatrix} \qquad (27)$$

For a more comprehensive understanding of the application of Lie groups in differential equations, we refer the reader to Olver (1993).

# B Proofs

## B.1 Lie Point Symmetry and Equivariance

**Proposition B.1.** *Let $G$ be a group that acts on $X$, the phase space of the ODE (1). $G$ is a symmetry group of the ODE (1) in terms of Definition 3.1 if and only if for any $\tau \in T$, the flow map $\boldsymbol{f}_\tau$ is equivariant to the $G$-action on $X$.*

*Proof.* The proof follows from applying various definitions. If $G$ is a symmetry group of (1), then for any solution $\boldsymbol{x} = \boldsymbol{x}(t)$, $\tilde{\boldsymbol{x}} = g \cdot \boldsymbol{x}$ is still a solution to (1). By definition of the flow map $\boldsymbol{f}_\tau$, we have $\boldsymbol{f}_\tau(\tilde{\boldsymbol{x}}_0) = \tilde{\boldsymbol{x}}_\tau$, where $\tilde{\boldsymbol{x}}_0 := g \cdot \boldsymbol{x}(t; \boldsymbol{x}_0)$ and $\tilde{\boldsymbol{x}}_\tau := g \cdot \boldsymbol{x}(t + \tau)$, for any $\boldsymbol{x}_0 \in X$. That is, $\boldsymbol{f}_\tau(g \cdot \boldsymbol{x}_0) = g \cdot \boldsymbol{x}(t + \tau) = g \cdot \boldsymbol{f}_\tau(\boldsymbol{x}_0)$, i.e. $\boldsymbol{f}_\tau$ is $G$-equivariant.

On the other hand, for any solution to the ODE (1) $\boldsymbol{x} = \boldsymbol{x}(t)$ where $\boldsymbol{x}(0) = \boldsymbol{x}_0$, we consider the transformed function $\tilde{\boldsymbol{x}} = g \cdot \boldsymbol{x}$. Because $\boldsymbol{f}_\tau$ is $G$-equivariant, we have

$$\tilde{\boldsymbol{x}}(t + \tau) = g \cdot \boldsymbol{x}(t + \tau) = g \cdot \boldsymbol{f}_\tau(\boldsymbol{x}(t)) = \boldsymbol{f}_\tau(\tilde{\boldsymbol{x}}(t)), \ \forall \tau \qquad (28)$$

Taking the derivative w.r.t $\tau$ at $\tau = 0$, we have $\frac{d}{dt}\tilde{\boldsymbol{x}}(t) = \boldsymbol{h}(\tilde{\boldsymbol{x}}(t))$, which means $\tilde{\boldsymbol{x}}$ is also a solution to the ODE (1). $\qquad \square$

**Theorem B.2.** *Let $G$ be a time-independent symmetry group of the ODE (1). Let $v \in \mathfrak{g}$ be an element in the Lie algebra $\mathfrak{g}$ of the group $G$. Consider the flow map $\boldsymbol{f}$ defined in (2) for a fixed time interval. For all $\boldsymbol{x} \in X$ and $g = \exp(\epsilon v) \in G$, the following equations hold:*

$$\boldsymbol{f}(g \cdot \boldsymbol{x}) - g \cdot \boldsymbol{f}(\boldsymbol{x}) = 0; \qquad (29) \qquad J_{\boldsymbol{f}}(\boldsymbol{x})v(\boldsymbol{x}) - v(\boldsymbol{f}(\boldsymbol{x})) = 0; \qquad (30)$$
$$J_g(\boldsymbol{x})\boldsymbol{h}(\boldsymbol{x}) - \boldsymbol{h}(g \cdot \boldsymbol{x}) = 0; \qquad (31) \qquad J_{\boldsymbol{h}}(\boldsymbol{x})v(\boldsymbol{x}) - J_v(\boldsymbol{x})\boldsymbol{h}(\boldsymbol{x}) = 0. \qquad (32)$$

*Proof.* From Proposition B.1, the flow map $\boldsymbol{f}$ is equivariant to $G$, so (29) follows directly.

Consider a group element in the identity component that can be written as $g = \exp(\epsilon v)$, $v \in \mathfrak{g}$. Taking the derivative with respect to $\epsilon$ on both sides of (29) at $\epsilon = 0$, we have

$$\frac{d}{d\epsilon}\left[\boldsymbol{f}(g \cdot \boldsymbol{x})\right]\Big|_{\epsilon=0} = \frac{d}{d\epsilon}\left[g \cdot \boldsymbol{f}(x)\right]\Big|_{\epsilon=0}$$
$$\Rightarrow \frac{d}{d\epsilon}\left[\boldsymbol{f}(g \cdot \boldsymbol{x})\right]\Big|_{\epsilon=0} = v(\boldsymbol{f}(\boldsymbol{x}))$$
$$\Rightarrow J_{\boldsymbol{f}}(\exp(0v)(\boldsymbol{x}))v(\boldsymbol{x}) = v(\boldsymbol{f}(\boldsymbol{x}))$$
$$\Rightarrow J_{\boldsymbol{f}}(\boldsymbol{x})v(\boldsymbol{x}) = v(\boldsymbol{f}(\boldsymbol{x}))$$

(31) and (32) can be proved similarly. Let $\boldsymbol{f}_\tau(\boldsymbol{x}_0) = \int_{t_0}^{t_0+\tau} \boldsymbol{h}(\boldsymbol{x})dt$ (where $\boldsymbol{x}(t_0) = \boldsymbol{x}_0$) be the flow map given the time interval $\tau$. Note that $\frac{d}{d\tau}|_{\tau=0}\boldsymbol{f}_\tau(\boldsymbol{x}) = \boldsymbol{h}(\boldsymbol{x})$. Since (29) is true for any $\tau$, we can take the derivative with respect to $\tau$ on (29) at $\tau = 0$ and obtain (31).

---

[3]Strictly speaking, the reverse only holds when $\Delta$ satisfy certain additional local solvability conditions.

Finally, we take the derivative with respect to $\tau$ on (30) at $\tau = 0$. We have

$$\big[\frac{d}{d\tau} J_{\boldsymbol{f}_\tau}(\boldsymbol{x})\big|_{\tau=0}\big]_{ij} = \frac{\partial}{\partial x_j}\frac{d}{d\tau}\big|_{\tau=0} f^i_\tau(\boldsymbol{x})$$

$$= [J_{\boldsymbol{h}}(\boldsymbol{x})]_{ij}$$

and

$$\frac{d}{d\tau} v(\boldsymbol{f}_\tau(\boldsymbol{x}))\big|_{\tau=0} = J_v(\boldsymbol{f}_{\tau=0}(\boldsymbol{x}))\boldsymbol{h}(\boldsymbol{x})$$

$$= J_v(\boldsymbol{x})\boldsymbol{h}(\boldsymbol{x})$$

which give us (32). $\qquad\square$

*Remark* B.3. (32) can also be obtained from the infinitesimal criterion (Theorem A.2). One can verify that when we assume time-independent symmetries and thus the prolonged vector fields given by (26), the symmetry condition in Theorem A.2 is equivalent to

$$\sum_j \frac{\partial \phi_i}{\partial x_j} x'_j = \sum_j \frac{\partial h_i}{\partial x_j}\phi_j, \quad i = 1, ..., d \qquad (33)$$

or, in its matrix form, $J_v(\boldsymbol{x})\boldsymbol{h}(\boldsymbol{x}) = J_{\boldsymbol{h}}(\boldsymbol{x})v(\boldsymbol{x})$, where $v(\boldsymbol{x}) = [\phi_1(\boldsymbol{x}), ..., \phi_d(\boldsymbol{x})]^T$.

*Remark* B.4. (30) is similar to some results in the existing literature, e.g. Section 4.2 in Finzi et al. (2021) and Theorem 3 in Otto et al. (2023). (30) is a generalization of these results which does not require the linearity of the function $\boldsymbol{f}$ or the infinitesimal action $v$.

## B.2 The Symbolic Map $M_\Theta$

In Section 4.1, we have defined the symbolic map $M_\Theta : (\mathbb{R}^d \to \mathbb{R}^n) \to \mathbb{R}^{n \times p}$, which maps each component of a function with multivariate output to its coordinate in the function space spanned by $\Theta$. This map is introduced to account for the fact that the constraint Proposition 4.1 holds for any $\boldsymbol{x}$. We apply this map to search for equal functions by their equal coordinates in the function space.

As an example, let the input dimension be $n = 2$ and $\Theta$ the set of all polynomials up to second order, i.e. $\Theta(x_1, x_2) = [1, x_1, x_2, x_1^2, x_1 x_2, x_2^2]^T$. Define $f_1(x_1, x_2) = [x_2^2, x_2]^T$ and $f_2(x_1, x_2) = x_1^3$.

We have $M_\Theta(f_1) = \begin{bmatrix} 0 & 0 & 0 & 0 & 0 & 1 \\ 0 & 0 & 1 & 0 & 0 & 0 \end{bmatrix}$, and $M_\Theta(f_2)$ undefined because the cubic term cannot be expressed in terms of a linear combination of functions in $\Theta$.

Thus, for a given infinitesimal action $v$, we need to ensure that an appropriate function library $\Theta$ is chosen for SINDy so that it can express the equivariant functions to the given action. For a linear action $v(\boldsymbol{x}) = L_v \boldsymbol{x}$, we prove that the polynomial functions satisfy the requirement.

**Proposition B.5.** *Let $\Theta(\boldsymbol{x})$ be the set of all polynomial functions up to degree $q$ ($q \in \mathbb{Z}^+$). Then, the components of $J_\Theta(\boldsymbol{x})L_v\boldsymbol{x} \in \mathbb{R}^p$ can be written as linear combinations of the terms in $\Theta(\boldsymbol{x})$, so that $M_\Theta(J_\Theta(\cdot)L_v(\cdot))$ is well-defined.*

*Proof.* Consider the function library $\Theta(\boldsymbol{x})$ that includes all polynomial terms up to degree $p \ge 1$. The Jacobian $J_\Theta(\boldsymbol{x})$ captures the partial derivatives of each component of $\Theta(\boldsymbol{x})$, which are polynomials up to degree $q - 1$. The linear transformation $L_v$ maps $\boldsymbol{x}$ to $L_v\boldsymbol{x}$, and therefore, is a first-degree polynomial in terms of $\boldsymbol{x}$. Hence, the product $J_\Theta(\boldsymbol{x}) \cdot L_v\boldsymbol{x}$ yields a function vector whose components are sums of products of two sets of polynomials: those of degree up to $q - 1$ from the Jacobian, and those of degree one from the linear transformation. The highest possible degree of any term in the product is $q$, as it is the sum of a polynomial of degree $q - 1$ and a polynomial of degree one.

Since $\Theta(\boldsymbol{x})$ includes all polynomials up to degree $p$, it includes all the terms generated by the product $J_\Theta(\boldsymbol{x}) \cdot L_v\boldsymbol{x}$. Therefore, the components of $J_\Theta(\boldsymbol{x})L_v\boldsymbol{x} \in \mathbb{R}^p$ can be written as linear combinations of the terms in $\Theta(\boldsymbol{x})$. $\qquad\square$

# C Supplementary Experimental Results

## C.1 Equivariant SINDy for Linear Symmetries

### C.1.1 Supplementary Results for Section 5.1

In addition to Table 1 in the main text, Table 3 shows the parameter estimation error and the long-term prediction error based on the discovered equations. For the parameter estimation error, we report the RMSE over successful runs and all runs. For the long-term prediction error, we select three temporal checkpoints and report the average prediction error of the equations at these checkpoints. The predictions from our discovered equations are much more accurate than equations from other methods. While our `EquivSINDy-c` achieves the best success probability and parameter estimation error over all runs, it is interesting to observe that Weak SINDy (WSINDy) has very accurate estimations of the constant parameters once it discovers the correct equation form (occasionally). We conjecture that this is because using the weak form of ODE averages out the white noise over a long time interval. Its low success probability suggests that the objective based on the weak form may introduce some difficulties in eliminating the incorrect terms. However, once it is given the correct form of the equation, parameter estimation can become much more accurate compared to other methods that use $L_2$ error at each individual data point with high measurement noise.

Table 3: Equation discovery statistics on the damped oscillator (13) at noise level $\sigma_R = 20\%$ and the growth system (14) at $\sigma_R = 5\%$. The success probability is computed from 100 runs for each algorithm. The success probabilities of recovering individual equations (Eq. 1 & Eq.2) and simultaneously recovering both equations (All) are reported. The RMSE (all) refers to the parameter estimation error over all runs. The RMSE (successful) refers to the parameter estimation error over successful runs, which is missing for algorithms with zero success probability.

| System | Method | Success prob. | | | RMSE (successful) ($\times 10^{-1}$) | | | RMSE (all) ($\times 10^{-1}$) | |
|---|---|---|---|---|---|---|---|---|---|
| | | Eq. 1 | Eq. 2 | All | Eq. 1 | Eq. 2 | All | Eq. 1 | Eq. 2 |
| Oscillator (13) | GP | 0.00 | 0.70 | 0.00 | N/A | 0.23 (0.11) | N/A | 0.71 (0.00) | 0.37 (0.24) |
| | D-CODE | 0.00 | 0.00 | 0.00 | N/A | N/A | N/A | 7.59 (9.35) | 7.53 (5.44) |
| | SINDy | 0.35 | 0.38 | 0.15 | 0.28 (0.06) | 0.35 (0.09) | **0.33** (0.04) | 0.50 (0.22) | 0.48 (0.21) |
| | WSINDy | 0.06 | 0.07 | 0.00 | **0.10** (0.05) | **0.09** (0.05) | N/A | 1.17 (0.56) | 1.07 (0.57) |
| | EquivSINDy-c | **0.93** | **0.97** | **0.90** | 0.37 (0.07) | 0.37 (0.08) | 0.37 (0.08) | **0.37** (0.08) | **0.37** (0.08) |
| Growth (14) | GP | 0.00 | **1.00** | 0.00 | N/A | 0.16 (0.14) | N/A | 2.13 (0.00) | 0.16 (0.14) |
| | D-CODE | 0.00 | 0.65 | 0.00 | N/A | 1.16 (0.28) | N/A | 2.17 (0.26) | 0.87 (0.55) |
| | SINDy | 0.26 | 0.13 | 0.03 | 0.20 (0.10) | **0.06** (0.05) | 0.22 (0.02) | 0.43 (0.51) | 0.50 (0.46) |
| | WSINDy | 0.00 | 0.00 | 0.00 | N/A | N/A | N/A | 2.45 (0.14) | 4.77 (3.34) |
| | EquivSINDy-c | **1.00** | **1.00** | **1.00** | **0.19** (0.10) | 0.08 (0.06) | **0.15** (0.07) | **0.19** (0.10) | **0.08** (0.06) |

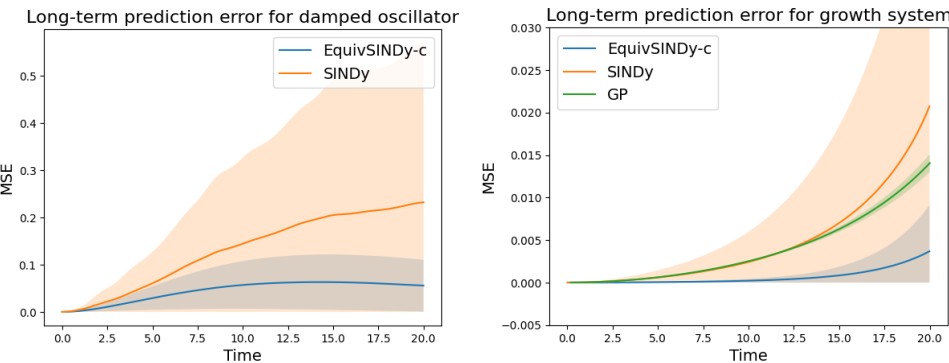

Figure 4: Long-term prediction error (MSE) against simulation time. The error curves are averaged over all discovered equations and random initial conditions in the test dataset. The shaded area indicates the standard deviation of prediction error at each timestep. Our `EquivSINDy-c` has the slowest error growth. Some algorithms, e.g. genetic programming and Weak SINDy, are not included in the plot when the simulation quickly diverges and the error grows to infinity.

Figure 4 plots the error growth curve when using the discovered ODEs to simulate future dynamics. Some algorithms, e.g. genetic programming and Weak SINDy, are not included in the plot when

the simulation quickly diverges and the error grows to infinity. We observe that the prediction error grows slowest when using the discovered equations from our method, `EquivSINDy-c`.

### C.1.2 A Higher-Dimensional System

Our method can also be applied to higher-dimensional dynamical systems with symmetries. To demonstrate this, we consider the SEIR equations Abou-Ismail (2020) which model the evolution of pandemics by the number of susceptible (S), exposed (E), infected (I), and recovered (R) individuals. It is a 4-dimensional ODE system with quadratic terms. We use the following equations as the target of equation discovery:

$$\begin{cases} \dot{S} = 0.15 - 0.6SI \\ \dot{E} = 0.6SI - E \\ \dot{I} = E - 0.5I \\ \dot{R} = -0.15 + 0.5I \end{cases} \tag{34}$$

Compared to the original formulation, we add two constant terms to the first and the fourth equation to account for the re-infection of patients due to waning immunity, mutation of the pathogen, etc. During data generation, we add a moderate $5\%$ noise to all trajectories.

We assume an intuitive symmetry of scaling the number of recovered individuals proportional to the total population: $v = (S + E + I + R)\partial_R$. We solve the constraint with regard to this symmetry as in Section 4.1. Figure 5 shows the reduced parameter space (from 60D to 34D).

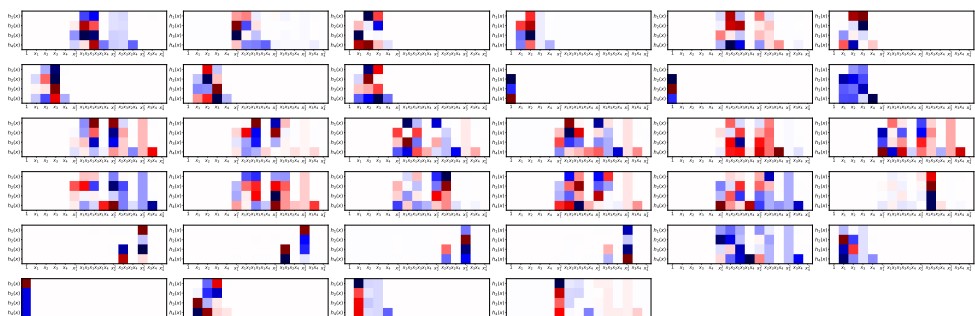

Figure 5: In the SEIR model, we search the equations of 4 variables with up to quadratic terms, leading to a 4×15=60D parameter space. The symmetry $v = (S + E + I + R)\partial_R$ reduces it to 34D.

Table 4 shows the discovery results of SINDy and our `EquivSINDy-c` in terms of success probabilities and parameter estimation error when successful. It turns out that SINDy struggles to find the correct equations and never manages to correctly identify all the equations simultaneously. In comparison, by reducing the parameter space through symmetry constraint, our method greatly increases the chance of successful discovery.

Table 4: Equation discovery on SEIR epidemic model (4D). Results are evaluated on 50 random runs.

| Dynamics | Method | Success prob. | | RMSE (successful) | |
|---|---|---|---|---|---|
| | | Individual eqs | Joint | Individual eqs | Joint |
| SEIR | SINDy | 0.02/0.02/0.20/0.10 | 0.00 | 1.57/**0.39/1.42/1.54** | - |
| | EquivSINDy-c | **0.26/0.98/0.84/0.64** | **0.14** | **1.22**/0.55/1.86/1.62 | **1.57** (0.47) |

### C.2 Symmetry Regularization

In addition to Table 2 in the main text, Figure 6 and Figure 7 show the error growth when using the discovered ODEs to simulate future dynamics. Some algorithms are not included in the plot because the simulation from the learned equation quickly diverges and the error grows to infinity. For the Lotka-Volterra system in Figure 6, we observe that the prediction error grows slowest when using the

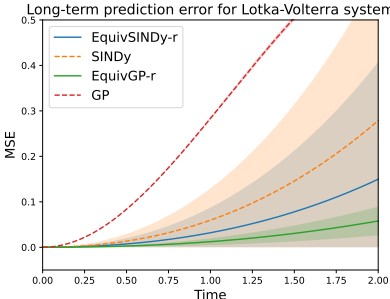

Figure 6: Long-term prediction error (MSE) on Lotka-Volterra system. The error curves are averaged over all discovered equations and random initial conditions in the test dataset. The shaded area indicates the standard deviation of prediction error at each timestep. Our methods based on symmetry regularization have slower error growth (visualized in solid lines).

discovered equations from our methods based on symmetry regularizations. Among all the methods, genetic programming with symmetry regularization has the most accurate prediction.

Figure 7 shows the results on the Sel'Kov glycolytic oscillator. Our symmetry regularization improves prediction accuracy over both of the base methods (SINDy and GP). However, D-CODE can achieve the best prediction accuracy in a long horizon, even though its discovered equations may not have the correct form.

Besides, it is noteworthy that genetic programming with symmetry regularization leads to a fluctuating error curve (after averaging). A potential reason is that it may discover some equations that are far from accurate but still lead to a periodic trajectory, similar to the true trajectory of the Sel'Kov glycolytic oscillator. The trajectory may have a large "eccentricity", so the error grows fast during a certain time and decreases rapidly afterward.

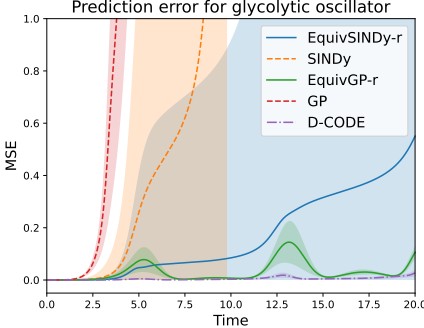

Figure 7: Long-term prediction error (MSE) on glycolytic oscillator. Left: results for sparse regression-based methods. Right: genetic programming-based methods. Our symmetry regularization improves prediction accuracy over the base methods. D-CODE can achieve the best prediction accuracy in a long horizon, even though its discovered equations may not have the correct form.

Besides, we also visualize the simulated trajectories of some learned equations in Figure 8. This provides a more intuitive explanation of the error growth. For example, the exploding variance in the long-term prediction error in the Lotka-Volterra system can be seen in Figure 8a, where predicted trajectories from most of the learned equations stay close to ground truth, but a few equations lead to diverging predictions due to the incorrect modeling of the exponential terms.

**Regularization Options.**    As mentioned in Section 4.2, we can introduce different forms of symmetry regularizations based on the four equations in Theorem 3.3 , listed as follows. The subscripts are

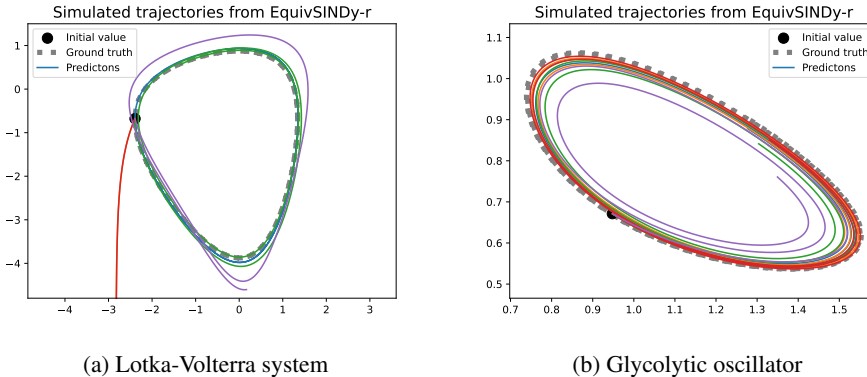

(a) Lotka-Volterra system          (b) Glycolytic oscillator

Figure 8: Simulated trajectories of the learned equations from `EquivSINDy-r`. In each figure, we sample 5 learned equations and start the simulation with the same initial value. The solid lines of different colors correspond to different discovered equations.

short for **I**nfinitesimal/**F**inite **G**roup actions and **I**nfinitesimal/**F**inite-time flow map of the OD**E**.

$$\mathcal{L}_{\text{IGFE}} = \mathbb{E}_{\boldsymbol{x}} \left[ \sum_{v \in B(\mathfrak{g})} \frac{\|J_{\boldsymbol{f}_\tau}(\boldsymbol{x})v(\boldsymbol{x}) - v(\boldsymbol{f}_\tau(\boldsymbol{x}))\|^2}{\|J_{\boldsymbol{f}_\tau}(\boldsymbol{x})v(\boldsymbol{x})\|^2} \right] \tag{35}$$

$$\mathcal{L}_{\text{FGFE}} = \mathbb{E}_{\boldsymbol{x}} \left[ \sum_{v \in B(\mathfrak{g})} \frac{\|\boldsymbol{f}_\tau(g \cdot \boldsymbol{x}) - g \cdot \boldsymbol{f}_\tau(\boldsymbol{x})\|^2}{\|\boldsymbol{f}_\tau(g \cdot \boldsymbol{x}) - \boldsymbol{f}_\tau(\boldsymbol{x})\|^2} \right] \tag{36}$$

$$\mathcal{L}_{\text{FGIE}} = \mathbb{E}_{\boldsymbol{x}} \left[ \sum_{v \in B(\mathfrak{g})} \frac{\|J_g(\boldsymbol{x})\boldsymbol{h}(\boldsymbol{x}) - \boldsymbol{h}(g \cdot \boldsymbol{x})\|^2}{\|J_g(\boldsymbol{x})\boldsymbol{h}(\boldsymbol{x})\|^2} \right] \tag{37}$$

$$\mathcal{L}_{\text{IGIE}} = \mathbb{E}_{\boldsymbol{x}} \left[ \sum_{v \in B(\mathfrak{g})} \frac{\|J_v(\boldsymbol{x})\boldsymbol{h}(\boldsymbol{x}) - J_{\boldsymbol{h}}(\boldsymbol{x})v(\boldsymbol{x})\|^2}{\|J_v(\boldsymbol{x})\boldsymbol{h}(\boldsymbol{x})\|^2} \right] \tag{38}$$

where $g = \exp(\epsilon v)$ is computed for some fixed scale parameter $\epsilon$. In other words, the summation is always performed over the learned infinitesimal generators ($v \in B(\mathfrak{g})$ and the group elements $g$ in the above formulas are dependent on $v$.

We also note that the denominator of (36) has a somewhat different form from others, i.e. the difference between $\boldsymbol{f}(g\boldsymbol{x})$ and $\boldsymbol{f}(\boldsymbol{x})$. If we use $\boldsymbol{f}(g\boldsymbol{x})$ or $\boldsymbol{f}(\boldsymbol{x})$ instead, the scale of the denominator would also depend on $\boldsymbol{x}$. As a result, regions closer to the origin in the state space have larger weights (smaller denominators) during loss computation. It's helpful to consider what would happen in the limit of $\epsilon \to 0, \tau \to 0$. The formulas inside the norms in the numerator and denominator in (35) both become $\boldsymbol{O}(\epsilon\tau)$, while $\boldsymbol{f}(g\boldsymbol{x}) = \boldsymbol{f}(\exp(\epsilon v)\boldsymbol{x}) = \boldsymbol{f}(\boldsymbol{x} + \boldsymbol{O}(\epsilon)) = \boldsymbol{x} + \boldsymbol{O}(\epsilon) + \boldsymbol{O}(\tau) + \boldsymbol{O}(\epsilon\tau)$, which does not match the numerator.

While the corresponding constraints in Theorem 3.3 of these losses are all necessary conditions for symmetry, these loss terms have different implementations that may bring certain advantages or disadvantages depending on the application, e.g. the representation of symmetry and the algorithm for equation discovery.

In practice, we use only one of the losses (35, 36, 37, 38) for training. Empirically, we find that (35, 36, 37) improve the discovery performance similarly while (38) leads to a performance drop. Detailed results are available in Table 5. We conjecture this is because we train a neural network to model the finite symmetry transformations (discussed in Section 4.3). Computing the Jacobian $J_v$ therefore involves computing second-order derivatives of the network and amplifies the numerical error. On the other hand, if we have exact knowledge about the symmetry of the system in terms of its infinitesimal generators $v$, (38) becomes the most efficient way to incorporate symmetry loss,

without the need of computing group elements based on the Lie algebra generators or integrating the learned ODE to its flow map.

Table 5: Equation discovery statistics of sparse regression with different regularization options (35, 36, 37, 38) on Lotka-Volterra system (15) at noise level $\sigma_R = 99\%$ and Sel'Kov oscillator (16) at noise level $\sigma_R = 20\%$.

| System | Regularization | Success prob. | | | RMSE (successful) | | | RMSE (all) | |
|---|---|---|---|---|---|---|---|---|---|
| | | Eq. 1 | Eq. 2 | All | Eq. 1 | Eq. 2 | All | Eq. 1 | Eq. 2 |
| L-V (15) | IGFE | 0.54 | **0.58** | 0.36 | 1.00 (0.21) | **0.45** (0.20) | **0.79** (0.15) | 3.16 (2.46) | 3.83 (4.01) |
| | FGFE | 0.54 | 0.56 | 0.34 | 0.99 (0.21) | **0.45** (0.20) | **0.79** (0.15) | 3.29 (2.65) | 4.00 (4.05) |
| | FGIE | **0.58** | **0.58** | **0.38** | 0.99 (0.20) | **0.44** (0.20) | **0.79** (0.14) | **2.90** (2.28) | **3.80** (4.00) |
| | IGIE | 0.48 | 0.16 | 0.08 | **0.58** (0.31) | 0.66 (0.17) | 0.88 (0.06) | 3.31 (2.67) | 7.61 (3.42) |
| Sel'Kov (16) | IGFE | 0.40 | **0.70** | **0.28** | 0.92 (0.22) | 0.30 (0.13) | 0.71 (0.16) | 9.97 (8.07) | **7.29** (12.72) |
| | FGFE | 0.48 | 0.68 | **0.28** | **0.87** (0.22) | **0.26** (0.12) | **0.63** (0.13) | 8.38 (8.55) | 7.97 (14.60) |
| | FGIE | 0.48 | 0.54 | **0.28** | 0.99 (0.19) | 0.72 (0.15) | 0.88 (0.08) | 8.79 (8.94) | 10.85 (17.50) |
| | IGIE | **0.50** | 0.48 | 0.22 | 0.94 (0.17) | 0.89 (0.16) | 0.91 (0.09) | **8.25** (8.09) | 14.56 (20.21) |

Besides, (37) and (38) have a specific advantage that any computation related to group actions is independent of the function $h$ which is constantly updated throughout training. Thus, we can compute the (infinitesimal) group actions and their Jacobians in advance over the entire dataset. This provides better compatibility between the use of symmetry and existing tools for equation discovery. For example, in PySR framework (Cranmer, 2023), it is difficult to compute group transformations $g \cdot x$ based on an external neural network module during the equation search. We overcome this challenge by pre-computing the transformations and using the transformed data as auxiliary inputs.

Table 6 summarizes the comparison between these losses. While their corresponding constraints in Theorem 3.3 are all necessary conditions for symmetry, they have different implementations that may bring certain advantages or disadvantages depending on the application, e.g. the representation of symmetry and the algorithm for equation discovery. For the experiments in Section 5.3, we use $\mathcal{L}_{\text{IGFE}}$ for sparse regression and $\mathcal{L}_{\text{FGIE}}$ for genetic programming.

Table 6: Comparison between different symmetry regularization losses on whether they need to compute (finite) group elements, compute higher-order derivatives (for infinitesimal action), integrate learned equations, and whether they can pre-compute the symmetry transformations over the data.

| Loss | (35) | (36) | (37) | (38) |
|---|---|---|---|---|
| Compute group elements | **No** | Yes | Yes | **No** |
| Higher-order derivatives | **No** | **No** | **No** | Yes |
| Integrate learned equations | Yes | Yes | **No** | **No** |
| Pre-compute symmetry | No | No | **Yes** | **Yes** |

### C.3 Equivariant SINDy in Latent Space

Table 7: Samples of discovered equations from SINDyAE Champion et al. (2019), LaLiGAN + SINDy Yang et al. (2023a) and our Equivariant SINDyAE.

| Method | Equation Samples | | |
|---|---|---|---|
| SINDyAE | $\dot{z}_1 = -0.98 + 0.16z_2^2$ | $\dot{z}_1 = 0.42z_2^2$ | $\dot{z}_1 = 1.08z_2 + 0.32z_2^2$ |
| | $\dot{z}_2 = 0.16z_2 + 0.80z_1^2$ | $\dot{z}_2 = 0.88z_1 + 0.37z_1^2 + 0.85z_2^2$ | $\dot{z}_2 = 0.28z_1$ |
| LaLiGAN+SINDy | $\dot{z}_1 = 0.10z_1 + 0.97z_2$ | $\dot{z}_1 = 0.99z_2$ | $\dot{z}_1 = 0.11z_1 + 0.92z_2$ |
| | $\dot{z}_2 = -0.88z_1$ | $\dot{z}_2 = -0.86z_1$ | $\dot{z}_2 = -0.92z_1 - 0.11z_2$ |
| EquivSINDyAE | $\dot{z}_1 = -0.88z_2$ | $\dot{z}_1 = -0.99z_2$ | $\dot{z}_1 = 0.78z_2$ |
| | $\dot{z}_2 = 0.92z_1$ | $\dot{z}_2 = 0.83z_1$ | $\dot{z}_2 = -1.04z_1$ |

In Table 7, we show some latent equations for the reaction-diffusion system in Section 5.2 discovered by different methods. Note that there is no single correct answer for the latent equations, because there might exist different latent spaces where the high-dimensional dynamics can be described accurately via different equations. Thus, the analysis in this subsection is mainly qualitative.

The discovered equations from SINDy Autoencoder (SINDyAE) contain numerous quadratic terms. While these equations may achieve a relative low SINDy error evaluated on individual timesteps, they fail to account for the periodic nature of the reaction-diffusion system. As a result, if we simulate

the system towards a long horizon with these equations, the error will accumulate rapidly, as shown in Figure 3. Also, the discovered equations from multiple runs are essentially different, making it difficult to select the best one to describe the dynamics.

Compared to SINDyAE, symmetry-based methods such as LaLiGAN + SINDy and our equivariant SINDy Autoencoder can stably discover simple linear equations with low prediction error. Moreover, we observe that our method yields more consistent results than the non-equivariant approach (LaLiGAN + SINDy). In LaLiGAN + SINDy, although the latent space has a linear symmetry action, the discovered equations are not constrained to satisfy the symmetry. The resulting equations can take slightly different forms, as shown in Table 7. On the other hand, our method constantly discovers equations in the form $\dot{z}_1 = -az_2$, $\dot{z}_2 = bz_1$, where the numerical relationship between $a$ and $b$ depends on the discovered symmetries.

### C.4 Comparison with Neural-ODE-Based Methods

We compare our method with the Time-Reversal Symmetry ODE Network (TRS-ODEN) (Huh et al., 2020), which uses the discrete symmetry of time-reversal to improve dynamics learning. TRS-ODEN aims to learn the physical dynamics with black-box neural networks and adds a loss term to encourage time-reversal symmetry. We evaluate TRS-ODEN as well as other baselines in Huh et al. (2020), i.e. Neural ODE (ODEN) and Hamiltonian Neural ODE (HODEN), on the damped oscillator system (13) and the Lotka-Volterra system (15) in terms of the long-term prediction error. Other metrics, including success probability and parameter estimation error, are not compared because Neural ODE does not learn a symbolic equation.

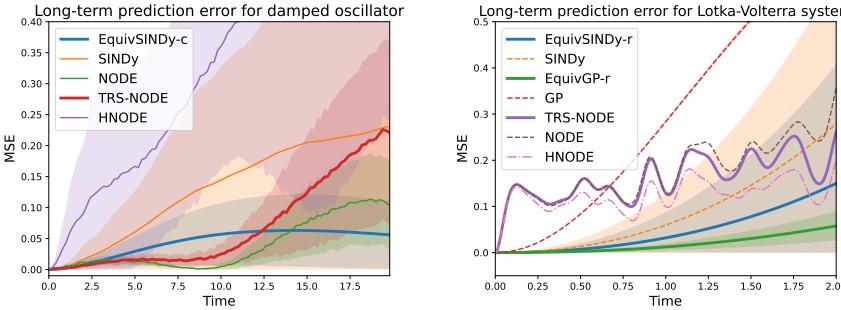

Figure 9: Long-term prediction errors of our method & Neural-ODE-based methods on the irreversible damped oscillator (left) and the reversible Lotka-Volterra system (right).

Figure 9 plots the long-term prediction error against the prediction horizon. In the damped oscillator, because it is irreversible and not conservative, TRS-ODEN and HODEN do not help. In the reversible Hamiltonian system of Lotka-Volterra equations, TRS-ODEN and HODEN improve upon the baseline Neural ODE, but have slightly higher errors than our discovered equations with Lie symmetry regularization. Notably, in the Lotka-Volterra system, our method does not rely on any prior knowledge (e.g. time-reversal symmetry or conservation of energy) but still achieves lower prediction error. To conclude, these results demonstrate that our method can be universally applied to various dynamical systems and achieve superior performance.

## D  Experiment Details

### D.1  Data Generation

**Damped Oscillator.**    We generate 50 trajectories for training, 10 for validation, and 10 for training, with $T = 100$ timesteps sampled at a fixed rate $\Delta t = 0.2$. The initial conditions are sampled by sampling the distance to origin $r \sim \mathcal{U}[0.5, 2]$ and the polar angle $\theta \sim \mathcal{U}[0, 2\pi]$ and then converting to $(x_1, x_2) = (r\cos\theta, r\sin\theta)$. The numerical integration is performed with a step size of $0.002$. We apply white noise with noise level $\sigma_R = 20\%$ to each dimension of the observation.

**Growth.** We generate 100 trajectories for training, 20 for validation, and 20 for testing, with $T = 100$ timesteps sampled at a fixed rate $\Delta t = 0.02$. The initial conditions are sampled uniformly from $[0.2, 1] \times [0.2, 1]$. The numerical integration is performed with a step size of $0.002$. Instead of applying white noise, we apply a multiplicative noise to this system, so the noise grows together with the scale of the state variables as the system evolves. Specifically, the observation is obtained by $\tilde{x}_i = x_i(1 + \epsilon_i)$, where $x_i$ is the true state from numerical integration and $\epsilon_i$ is sampled from a Gaussian distribution with zero mean and standard deviation $\sigma = 0.05$. We still refer to this as $5\%$ noise level in the main paper. While the assumption of Gaussian process does not hold here, we find that Gaussian process smoothing can still improve the likelihood for the equation discovery algorithms to recover the correct equation. Therefore, just as in any other ODE systems considered in this work, we apply Gaussian process smoothing before running equation discovery methods.

**Lotka-Volterra System.** We generate 200 trajectories for training, 20 for validation, and 20 for testing, with $T = 10000$ timesteps sampled at a fixed rate $\Delta t = 0.002$. For initial conditions, we first sample the population densities $(p_1, p_2)$ uniformly from $[0, 1] \times [0, 1]$. Then they are converted to the logarithm population densities in the canonical form of the equation, i.e. $x_1 = \log(p_1)$ and $x_2 = \log(p_2)$. Also, we ensure that the Hamiltonian of the system $\mathcal{H} = \exp(x_1) - x_1 + 1.333 \exp(x_2) - 0.667$ under the sampled initial conditions falls within the range of $[3, 4.5]$. We discard the initial conditions that do not meet this requirement. Then, the numerical integration is performed with a step size of $0.002$. We apply white noise with noise level $\sigma_R = 99\%$ to each dimension of the observation.

**Glycolytic Oscillator.** We generate 10 trajectories for training, 2 for validation, and 2 for testing, with $T = 10000$ timesteps sampled at a fixed rate $\Delta t = 0.002$. The initial conditions are sampled uniformly from $[0.5, 1] \times [0.5, 1]$. The numerical integration is performed with a step size of $0.002$. We apply white noise with noise level $\sigma_R = 20\%$ to each dimension of the observation. We note that our experimental results on this system are somewhat different from the results in Qian et al. (2022) on this system with the same noise level. This is because our initial conditions are sampled from a larger range which leads to more diverse trajectories. As a result, the metrics such as success probabilities and parameter estimation errors appear worse than in Qian et al. (2022). However, we still ensure fair comparison as we use the same generated dataset with our configurations for all the methods involved in our experiment.

**Reaction-Diffusion System.** The system is governed by the PDE
$$u_t = (1 - (u^2 + v^2))u + (u^2 + v^2)v + 0.1(u_{xx} + u_{yy}),$$
$$v_t = -(u^2 + v^2)u + (1 - (u^2 + v^2))v + 0.1(u_{xx} + u_{yy}).$$

We use the code from SINDy Autoencoder [4] (Champion et al., 2019) to generate the data. The 2D space of $u$ and $v$ is discretized into a $100 \times 100$ grid. The system is simulated from one initial value for $T = 6000$ timesteps with step size $\Delta t = 0.05$. Then, we add random Gaussian noises with standard deviation $10^{-6}$ to each pixel and at each timestep. We use the timesteps $t \in [0, 4800]$ for training. For forecasting, we use the timestep $t = 4800$ as the initial value and simulate up to 600 timesteps with each method. The simulations are compared with the ground truth during $t \in [4800, 5400)$ to calculate the error.

### D.2 Equation Discovery

#### D.2.1 Sparse Regression

For all algorithms based on sparse regression (SINDy), we use a function library $\Theta(\boldsymbol{x})$ that contains up to second-order polynomials, with an exception for Lotka-Volterra system where we also include the exponential terms. We apply sequential thresholding (Brunton et al., 2016) to enforce parsimony in the equations. The threshold is set to $0.05$ for the damped oscillator and the growth system, $0.075$ for the glycolytic oscillator, and $0.15$ for the Lotka-Volterra system. The same threshold is applied to all methods including SINDy, WSINDy and `EquivSINDy`.

The original SINDy uses a least square objective and can be solved explicitly. However, when we introduce symmetry regularization into the training objective, the original solver is no longer

---

[4] https://github.com/kpchamp/SindyAutoencoders/tree/master/rd_solver

applicable. For our equivariant models based on symmetry regularization, we use the L-BFGS algorithm (Nocedal, 1980) for optimizing the SINDy parameters. We find L-BFGS more effective than stochastic gradient descent for these low-dimensional problems.

### D.2.2 Genetic Programming

We incorporate our symmetry regularization (37) to an existing genetic programming package for discovering equations, PySR (Cranmer, 2023). The "GP" baseline in the tables also refers to the implementation of genetic programming with PySR. On the other hand, we run the experiments on D-CODE with their official codebase [5]. This codebase is developed based on another genetic programming package, gplearn. We should note that the performance difference between our EquivGP and D-CODE can be partially caused by their different backends.

---

[5] https://github.com/ZhaozhiQIAN/D-CODE-ICLR-2022

