# OpenReview forum: "Symmetry-Informed Governing Equation Discovery"
_NeurIPS.cc/2024/Conference — NeurIPS 2024 poster_

### Official Review · Reviewer_CnKF · 2024-07-01

**Soundness:** 3
**Presentation:** 2
**Contribution:** 3
**Rating:** 7
**Confidence:** 2

**Summary:**

The authors developed an approach that allows exploitation of symmetry in common equation discovery algorithms and test their approach on dynamic systems with and without symmetry.
For this they make use of Lie groups.

**Strengths:**

- Despite Lie groups being a new topic for me, I was able to follow the well written high-level explanations in Sections 3.1 and 3.2
- I like the approach that was used to exploit additional structure in the search
- The results are very strong on the chosen systems, though I am still left sceptic of _how much_ the results are better in Table 1.
- The experiments section is written concisely.

**Weaknesses:**

_Weaknesses_
- Just looking at Figure 1, I find it hard to recognize at which points the dynamics are searched, i.e. where the equation discovery itself is happening. From the text prior to Figure 1, I expect the authors approach to find the symmetry first, constrain the search space and then perform equation discovery, but Figure 1 suggests otherwise.
- Chapter 4 could benefit from examples (or a running example) how corresponding functions might look like, to facilitate understanding. In particular since $M_\Theta$ can be computed in a symbolic fashion. Since the work is already tightly packed, such examples can also be mentioned in the appendix.


_Minor complaints/typos_
- A small arrow/text indicating the start "Start" in the flowchart on the left side of Figure 1 would make orientation slightly easier.
- Section 4 was harder to read, compared to section 3
- 261: With RK4 you mean Runge-Kutta 4 methods?
- Figure 2 could be bigger to improve readability.



To summarize: I enjoyed this work and find it relevant to the community and a good contribution. However, the explanations in Chapter 4 were partly unclear to me and could benefit from an additional revision.
The most conflicting point for me are the results of Table 2, which I hope to have clarified during the discussion.

**Questions:**

I want to highlight a major question before all others.
With respect to the results in Table 2:
- If there is no symmetry in the system, shouldn't your approaches fallback to their respective base approaches, i.e. SINDY and GP? Why does your approach still outperform them so consistently, despite there being no exploitable symmetry in the ODEs?
- Did you look into the results of the LaLiGAN? Did you recover the non-existence of symmetries and _still_ performed better? Or did it find some symmetry which your approach then used?

Other questions:

- 78/79: Thank you for highlighting works that already exploit symmetries. Can you provide a brief description of how you differ from Loiseau & Brunton 2018 and Guan et al. 2021?
- I have a minor question for Prop 3.2. What does it mean for the flow map  $f_\tau$ to be equivariant to the G-action on X? Does that mean that $f_\tau$ needs to still be a valid flow map after an element of the symmetry group has been applied to $X$?
- For Theorem 3.3 I have a few minor questions:
	- What is $\epsilon$  in the $g = \exp(\epsilon v)$ term?
	- Why is $g$ defined this way? Would it not be possible to calculate the Jacobian otherwise?
	- Is it guaranteed that $g \in G$? Does that only hold because $G$ is a symmetry group? Or generally?
	- I assume that $v$ is a matrix, please correct me if I am wrong, what does it mean to perform $\exp(\epsilon v)$ in that case? Is this the matrix exponential?
- What is $L_v$ in Proposition 4.1? Is it a linear transformation?
- 164/165: I don't understand what $c$ is and what its purpose is
- 180: The set of polynomials doesn't include products $x_1 x_2$ like in the example for SINDY in line 163, correct?
- 183: Is $W$ vectorized with respect to rows $\{[w_{11} ... w_{1n}], ...\}$  or columns $\{[w_{11} ... w_{n1}], ...\}$?
- Where does the factorization in Equation (10) come from? Are there established methods that easily generate this? What do its components mean?

**Limitations:**

In my opinion the only limitation of this work lies in the experiments in 5.3, as I stated before.
I hope the authors can either point out how their approach differs from vanilla SINDy/GP and explain why it is better or demonstrate, perhaps using a "neutral" operator instead of LaLiGAN, that it does fall back and that the difference is due to LaLiGAN.

---

> ### Author Rebuttal · Authors · 2024-08-01
>
> Thank you for the valuable feedback. We address key comments below, starting with the major question about Tab. 2.
> > *If there is no symmetry*, why does our approach still outperform baselines?
>
> There *are* symmetries in the tasks in Tab. 2. If there were no symmetry, our approach would not outperform baselines. The confusion might arise because we mentioned “these systems do not possess any *linear* symmetries” in L348-349. But they do have *non-linear* symmetries. For the Lotka-Volterra system, LaLiGAN learned a “distorted” rotation symmetry in their paper. We used LaLiGAN and obtained a similar symmetry, which we then used for regularization.
>
> **This should also address the concerns in the Limitations part**. Please let us know if you have other questions about this point.
> > Understanding Fig. 1
>
> Your understanding of the text is correct. In Fig. 1, equation discovery is performed in the last column (green background, titled “learning”). The confusion might have arisen because we titled the first column “dynamics”, where we actually meant the data collected from dynamical systems. We will change the block titles to avoid potential confusion. Thanks for pointing this out!
> > Example of computing $M_\Theta$
>
> We had an example in Appendix B.2 (L605-608). We further explain it here.
>
> Define $\Theta(x_1,x_2)=[1,x_1,x_2,x_1^2,x_1x_2,x_2^2]^T$, and $f(x_1,x_2) = [x_2^2,x_2]^T$. Each row of $M_\Theta(f)$ corresponds to one component of $f$. For the first row, we look at $f_1(x_1,x_2)=x_2^2$. It is the linear combination of the elements in $\Theta$ with coefficients $[0,0,0,0,0,1]$, which becomes the first row of $M_\Theta(f)$. Similarly, the second row is $[0,0,1,0,0,0]$.
> > On our difference from related works (Loiseau & Brunton 2018 and Guan et al. 2021)
>
> Our work addresses a broad range of symmetries in dynamical systems; our pipeline is also applicable when we do not know the symmetry a priori. These related works, however, only considered a few instances of symmetries based on prior knowledge about specific systems. E.g. Loiseau (2018) only considered the time translation symmetry (from the energy-preserving principle) of a reduced-order modeling of Navier-Stokes equations. Guan (2021) considered a set of reflection and permutation symmetries of the proper orthogonal decomposition (POD) coefficients.
> > Other questions
>
> The reviewer raised questions about specific details in the paper. We believe all related contents in our paper are mathematically correct, but agree some explanation would be helpful. We answer the questions below and will add necessary details to the updated version.
> * L261: yes, we mean the Runge Kutta 4 method by RK4.
> * Prop 3.2: $f_\tau$ is equivariant to $G$-action on $X$, if $f_\tau(g.x)=g.f_\tau(x),\ \forall g,x$. This is the definition of equivariance. It is somewhat inaccurate to state that “$f_\tau$ is still a valid flow map after a group element is applied”. The group element $g$ does not transform the function $f_\tau$ itself, but its input and output $x, f_\tau(x) \in X$.
> * Thm 3.3: this involves some knowledge of Lie's theory. We refer to Appendix A.1 for background knowledge and answer the specific questions below.
>     * $\epsilon\in R$ is the scaling factor that determines the “size” of the transformation $g$.
>     * $g$ is defined as such because we consider the Lie group of continuous transformations. It is not related to the Jacobian computation. If we have a Lie algebra element $v$ (or $\epsilon v$, which is still in the Lie algebra), we can always get a group element $g\in G$ by applying the exponential map.
>     * (Is it guaranteed that $g\in G$?) Yes, it follows from definitions of Lie group and Lie algebra. See above.
>     * (Does $g\in G$ hold because $G$ is a symmetry group?) No, it holds for a general Lie group. This is how we describe a Lie group element: by its counterpart in the Lie algebra, a flat vector space that is easier to deal with. At this point, we don’t need to consider whether $G$ is the symmetry of ODE. We just consider $G$ as a Lie group itself.
>     * (Is $v$ a matrix? What is $\exp$?) $v$ could be a matrix, but not necessarily. More generally, $v$ could be a nonlinear vector field, and $\exp$ can be understood as following the flow generated by that vector field. Intuitively, think of a 2D plane. The vector field $v: R^2 \to R^2$ consists of arrows pointing in various directions at every point. Exponentiating the vector field is like starting at one point and walking in the arrow directions for a certain time $\epsilon$. The path you take following these arrows traces your journey, and the ending point represents the effect of the vector field exponentiation. In terms of computation, if $v$ is a general vector field, $\exp$ is computed by integrating the vector field for a certain time interval; if $v$ is a matrix, it is equivalent to matrix exponential.
> * Prop 4.1: yes, $L_v$ is a matrix, and $L_v x$ is mat-vec multiplication.
> * L164-165 (what is $c$?): it reads as: if a function $f$ is a linear combination of the $p$ functions in $\Theta$, with the coefficients $c\in R^p$, i.e. $f(x)=c^T\Theta(x)$, then $M_\Theta(f)$ is defined as $c$.
> * L180 (does the set of polynomials include $x_1x_2$?): Yes, it includes products of different $x_i$’s. Polynomials of multiple variables include terms like $x_1^{q_1}...x_d^{q_d}$, where the polynomial degree is defined as $q = \sum_i q_i$.
> * L183: As in common practice, matrix vectorization stacks the columns: $\mathrm{vec}(W)=[w_{11},...,w_{d1},w_{12},...]$.
> * Eq (10) refers to singular value decomposition. There are built-in SVD functions in any modern numerical computation package (e.g. torch.svd). Of particular interest is $Q^T$, the right singular vectors corresponding to zero singular values. They span the null space of $C$, i.e. the solution space of $W$.
>
> Please let us know if you have other questions. If your concerns have been addressed, may we kindly request you increase the score?

---

> > ### Comment · Reviewer_CnKF · 2024-08-09
> >
> > Thank you for answering my questions and providing clarifications.
> > I happily updated my score.

---

### Official Review · Reviewer_HCHE · 2024-07-02

**Soundness:** 3
**Presentation:** 4
**Contribution:** 3
**Rating:** 7
**Confidence:** 3

**Summary:**

This paper propose to leverage the symmetry to discover underlying dynamics (especially the ones described by autonomous ODEs) correctly from data. Specifically, the proposed method combines conventional symbolic regression approaches like SINDy with the symmetry-based constraints or regularizations. The used symmetry-based regularization is derived from the Lie group symmetry, namely the flow of the given dynamics should be $G$-equivariant with respect to the given Lie group $G$ of symmetries. For the case that the group of symmetry transformations is unknown, the authors use some recent techniques that can learn the unknown symmetry from data, and parameterize the Lie algebra of the learned symmetry with neural networks. In this case, the extracted symmetry regularization can be encoded by using the infinitesimal version of $G$-equivariance with the learned Lie algebra generators. The authors validate their proposed approach with some dynamical systems.

**Strengths:**

**Significance.** Using symmetries to generalize the machine learning dynamics is very relevant topic. The combination of the symbolic regression with the symmetry regularization for recovering the underlying dynamics is thus very principle.

***

**Clarity.** The paper is well-written and easy to follow. The main paper is concise and highlights the advantages of the proposed technique without heavy math/details. Readers who want to know the detailed background and techniques on the proposed method can find the details in the appendix.

**Weaknesses:**

**Novelty.** The proposed technique is straightforward, if one already knows the group of symmetry transformations that should be applied. It is basically the regularization for the (infinitesimal) $G$-equivariance of the given Lie group $G$, i.e., $\lVert f(g \* x) - g \* f(x) \rVert$. Therefore, the interesting part of the proposed method is the case that the symmetry is unknown. In this case, the authors rely on [1], which can find the infinitesimal generators of Lie group of symmetries from data. Because they use [1] without clear modification/enhancements, there is some doubt as to whether the combination of two already known techniques (equivariance regularization and symmetry discovery) can be sufficiently novel.

[1] Yang et al. Latent Space Symmetry Discovery. arXiv 2023.

***

**Experiments.** The proposed method is evaluated on relatively simple dynamics. It would be beneficial if the authors could provide experimental results on more complicated dynamics, such as chaotic systems.

***


There are some typos, e.g., the denominator of (35), and the summation spaces ($v$ instead of $g$) of (35) and (36)

**Questions:**

The author mentions that though there are four forms of Lie point symmetry (i.e., equations (4 – 7) or equations (34 – 17)), they are empirically similar in terms of the performance. What would happen if the authors use multiple ones simultaneously instead of just one of these four? For example, how would the performance be if you use both (34) and (35) at the same time? I think this would work similarly to the Sobolev training [2], thus potentially improving performance, especially when the dataset is noisy.

[2] Czarnecki et al. Sobolev Training for Neural Networks. NeurIPS 2017.

**Limitations:**

The authors mention that their method only consider the time-independent point symmeties for autonomous ODEs. They provides some potential future research directions for generalizing their approaches, e.g., for non-autonomous ODEs.

---

> ### Author Rebuttal · Authors · 2024-08-05
>
> We thank the reviewer for the valuable feedback and the recognition of our paper's significance and clarity. We address key comments below.
>
> > Novelty
>
> Our method is *not* a combination of existing techniques.
>
> First, we develop the pipeline for using different kinds of symmetries in equation discovery. For a known linear symmetry, we propose to solve the symmetry constraint explicitly with Lie’s infinitesimal criterion (Section 4.1), which is our novel contribution.
>
> Then, when the symmetry is unknown, we use symmetry discovery techniques to learn it from data. It does not have to be LaLiGAN, but we choose it in our experiments because of its ability to express nonlinear symmetries. We also make important adaptations to incorporate the discovered symmetry into our approach. E.g. We extract the infinitesimal action in the original state space as in eq (12), which is essential for symmetry regularization. This is not discussed in LaLiGAN (which only addressed linear infinitesimal actions in latent space); our approach opens up new possibilities for applying the learned symmetries. We also proposed the relative losses for regularization to prevent bias in the learned equations. An ablation study for relative loss is provided in the supplementary pdf (Table 1). We show that relative loss improves success probability and reduces parameter estimation errors on Lotka-Volterra equations.
>
> > Experiment with more complicated dynamics
>
> In the supplementary PDF (**Table 4**), we provide two examples of learning higher-dimensional dynamics by solving the hard constraint (SEIR) and symmetry regularization (Lorenz).
>
> The SEIR equations [1] model the evolution of pandemics by the number of susceptible, exposed, infected, and recovered individuals. It is a 4-dimensional ODE system with quadratic terms. We consider the following equations:
>
> $
> \\left\\{
> \begin{aligned}
> \dot S &= 0.15-0.6SI\\\\
> \dot E &= 0.6SI-E\\\\
> \dot I &= E-0.5I\\\\
> \dot R &= -0.15+0.5I
> \end{aligned}
> \\right.
> $
>
> We assume an intuitive symmetry of scaling the number of recovered individuals proportional to the total population: $v=(S+E+I+R)\partial_R$. We solve the constraint w.r.t this symmetry as in Sec 4.1. **Figure 2** (supplementary PDF) shows the reduced parameter space (from 60D to 34D).
>
> For the 3D Lorenz system [2], we use parameters $\sigma=0.5,\beta=1.0,\rho=0.0$, leading to non-chaotic dynamics. Similar to Section 5.3, we discover the symmetry first and use the learned symmetry to regularize equation discovery.
>
> The presence of a nontrivial Lie symmetry often implies some form of simplification or integrability [3], which contradicts the nature of chaotic systems. We are not aware of any nontrivial Lie symmetry in chaotic dynamics.
>
> [1] Abou-Ismail A. Compartmental Models of the COVID-19 Pandemic for Physicians and Physician-Scientists. SN Compr Clin Med. 2020;2(7):852-858.
>
> [2] Brunton et al. Discovering governing equations from data by sparse identification of nonlinear dynamical systems. Proceedings of the National Academy of Sciences, 2016
>
> [3] Olver. Applications of Lie groups to differential equations. Springer Science & Business Media, 1993.
>
> > Equations (34-37)
>
> Thank you for noticing these! These are **not** typos but do require more clarification. The summation spaces should indeed be $v \in B(\mathfrak g)$, because we can only enumerate the finite list of Lie algebra basis components, not the infinite group elements. The group elements in (35)(36) are obtained from $v$ by $g=\exp(\epsilon v)$, where $\epsilon$ is chosen manually. We can make the dependency of $g$ on $v$ explicit to prevent confusion.
>
> Also, the denominator of (35) should indeed be the difference between $f(gx)$ and $f(x)$. If we use $f(gx)$ instead, the scale of the denominator would also depend on $x$. As a result, regions closer to the origin in the state space have larger weights (smaller denominators) during loss computation. It’s helpful to consider what would happen in the limit of $\epsilon \to 0, \tau \to 0$. The formulas inside the norms in the numerator and denominator in (35) both become $O(\epsilon\tau)$, while $f(gx) = f(\exp(\epsilon v)x) = f(x+O(\epsilon)) = x+O(\epsilon)+O(\tau)+O(\epsilon\tau)$, which does not match the numerator.
>
> > Sobolev training
>
> Thank you for the reference. The idea of Sobolev training is indeed relevant in this case. In the supplementary pdf (**Table 3**), we did additional experiments on both systems in Section 5.3, where we tried the combinations of (34)+(35) and (35)+(36). The latter involves $h$ and $f=\mathrm{odeint}(h)$, which is more similar to the formulation of Sobolev training because we are learning the dynamics $h$ instead of the group action $v$. We compared them with the best results from using a single loss term. The combination (35)+(36) has a slightly higher success probability of finding the correct equation forms than the original results using single losses on the Lotka-Volterra system. For reference, the full original results are in Table 5 in the paper.

---

> > ### Comment · Reviewer_HCHE · 2024-08-12
> >
> > Thank you for the authors' thoughtful response. Most of my concerns have been resolved, so I will be raising the review score.

---

### Official Review · Reviewer_zT4E · 2024-07-07

**Soundness:** 3
**Presentation:** 3
**Contribution:** 2
**Rating:** 5
**Confidence:** 3

**Summary:**

The authors consider an estimation technique similar to the well known SINDy technique for recovering interpretable parameterizations of ordinary differential equations (ODEs). The authors primarily consider the context where there exists a Lie symmetry that constrains the solution space of ODEs to search over. The authors describe different ways in which to incorporate the symmetry information within the learning problem. The first is to consider linear constraints in which case the problem simplifies into another coefficient regression optimization. The second involves general constraints for which the authors propose a regularization scheme to constrain the function class. These methods are described in terms of both the case of observed data and a learned coordinate system. Finally, the authors consider a few different experiments where they try to estimate the coefficients for known data as well as predict the evolution of the data forward in time.

**Strengths:**

Including the symmetry information is a natural approach to try to constrain the learning problem into something more manageable. The general SINDy learning problem can be difficult to interpret due to the requirement of the specification of the basis. In that sense, this method provides a good solution for imposing some constraints on the function space to make the problem more tractable.

The numerical results suggest the method performs very well compared to the baselines. For the forward prediction on the reaction diffusion system, the method also performs better than the related baselines. This shows the promise in some more general tasks such as time series forecasting  given sufficient regularity on the observed data.

**Weaknesses:**

The method largely focuses on cases where symmetries are linear. The regularization approach in equation 11 seems a bit arbitrary in its formulation (e.g. using the relative error) as well as it seems like the results are primarily applicable in the cases where the symmetry is linear.

Some of the empirical results are slightly confusing, in particular when the baselines fail catastrophically under some circumstances leading to high RMSE. However, in the cases where the the baselines successfully identify the equation, the baselines tend to have smaller RMSE. This may be a hyperparameter tuning issue, but it would seem more intuitive that the method with the symmetry should be easier to achieve a lower RMSE.

**Questions:**

Is there any existing theory on how well learned the symmetries can be from the data?

How does the computational cost of the method compare? I'm particularly interested in the increase in cost associated with the vector Jacobian product computation.

Is there any intuition why the RMSE of W-SINDy is better than the proposed method in the case of successful identification?

**Limitations:**

The authors briefly discuss limitations regarding requiring known symmetries for the algorithm. It should probably be discussed more on how performance changes when using prior knowledge versus when learning these symmetries from the data.

---

> ### Author Rebuttal · Authors · 2024-07-31
>
> We thank the reviewer for the valuable feedback. We address key comments below.
>
> > The method focuses on cases where symmetries are linear. The results are primarily applicable in the cases where the symmetry is linear.
>
> **This is a misunderstanding.** Our regularization method in Section 4.2 applies to general Lie symmetries, including linear and nonlinear ones. Specifically, the infinitesimal symmetry $v$ in eq. 11 can be either linear or nonlinear functions. Also, experimental results in Section 5.3 show that our method can work with nonlinear symmetry. Please let us know if you have additional questions regarding this point.
>
> > The relative error in eq. 11 seems arbitrary in its formulation
>
> Using relative error in eq 11 is not arbitrary. We have explained the rationale for this formulation in the text following eq 11. The main idea is that an absolute loss would decrease with the magnitude of $h$. If we used the absolute error instead of the relative error, it would bias the discovered $h$ (towards a lower magnitude) and lead to a large coefficient estimation error.
>
> To demonstrate this, we provide an ablation study on the Lotka-Volterra system, using both relative error and absolute error. From the table below, the absolute error causes a larger negative bias (computed as the mean of $(\hat \theta_i - \theta_i) / \theta_i$), meaning the discovered coefficients tend to have smaller magnitudes. It also increases the RMSE and reduces the success probability of finding the correct equation form. For reference, the results in the “None” and “Relative” rows correspond to Table 4, “L-V: SINDy” and “L-V: EquivSINDy-r” rows, in the paper.
>
> > Interpretation of empirical results, in particular when baselines achieve lower RMSE
>
> Our symmetry-based methods mainly increase the **success probability** of finding correct equation forms, which is the primary metric as we discussed in L275-277. This demonstrates the main benefits of using symmetries.
>
> Symmetry-based methods can have higher parameter estimation RMSE when the symmetry learned from data is not perfectly accurate. In comparison, when we know the exact symmetry in Section 5.1, the RMSEs of our method and SINDy w/o symmetry are similar.
>
> To understand this, note that symmetry improves equation discovery by identifying a subspace of possible equations. When we explicitly solve the constraint (Sec 4.1), we enforce the model to search strictly within this space; when we use regularization (Sec 4.2), we encourage it to search around it. In either case, this smaller subspace makes it more likely to identify the correct solution. This explains the higher success probability of symmetry-informed methods. On the other hand, once the model (either with or without symmetry) manages to reach the proximity of the correct solution (i.e. it has found the correct equation form), symmetry no longer helps. At this stage, a slightly inaccurate symmetry learned from data may even cause a small bias, which explains why our symmetry-based method in Sec 5.2 sometimes has a higher RMSE. **Figure 1** in the supplementary PDF demonstrates the above argument with the equation space abstracted into a 2D plane.
>
> Moreover, to reduce the influence of inaccurate learned symmetry, we can refine the discovered equation w/o regularization. Specifically, we first perform the same experiment as in the paper with symmetry regularization. Then, we remove the regularization, fix the form of the equation, and fit the equation parameters under the equation fitting loss (same as baseline). The table shows that this refinement process can further reduce the parameter estimation error on the glycolytic oscillator. For reference, the “SINDy” and “EquivSINDy-r” rows here correspond to the “Sel’Kov” rows and RMSE (successful) columns in Table 4 of the paper.
>
> Please let us know if the above discussions clear up the confusion. We will incorporate these comments in the revision.
>
> > Any existing theory on how well learned the symmetries can be?
>
> That is an interesting question! To our knowledge, there isn’t such a theory that bounds the error of the learned symmetry in the latest works on symmetry discovery. Some relevant theoretical results include the necessary conditions for symmetry discovery with GAN [1], and quantitative metrics for learned symmetry [2,3]. We will leave this to our future work.
>
> [1] Generative Adversarial Symmetry Discovery. ICML 2023.
>
> [2] Latent Space Symmetry Discovery. ICML 2024.
>
> [3] LieGG: Studying Learned Lie Group Generators. NeurIPS 2022.
>
> > Computational cost of our method
>
> Our method moderately increases the computational cost. Excluding common procedures across methods, e.g. data loading, SINDy has an average run time of 11.62s and ours has an average of 20.44s over 50 runs on the L-V system, adding only <1x computational overhead. The computational cost also depends on how complicated the symmetry is. E.g. if we use an arbitrary simple symbolic function as symmetry (which certainly affects accuracy, but it’s fine since we are investigating efficiency), instead of the 5-layer MLPs in LaLiGAN, the average running time decreases to 13.07s, because the JVP computation is cheaper.
>
> > Any intuition why the RMSE of W-SINDy is better?
>
> We have discussed this briefly in Appendix C.1 (L633-639). The intuition is that WSINDy computes loss based on the integration within a time interval, instead of time derivative errors at individual points. The white noise is averaged out over this time interval. As a discrete analogy, let $X_1,...,X_n$ be i.i.d standard Gaussian RVs, then their mean has reduced variance $1/n$. This explains why WSINDy is better at parameter estimation with our highly noisy data.

---

> > ### Comment · Reviewer_zT4E · 2024-08-11
> >
> > Many thanks to the authors for the response, I have read and thought about it. On the linear comment, most of interesting parts of the paper are devoted to the linear case, which makes sense since this can be more tractably analyzed. The nonlinear case and its relative error regularization loss appear to be an afterthought. This is mainly because this regularization does not provide a hard constraint that the linear case does; rather, the symmetry is only satisfied to the extent that the regularization term is minimized (similar to a PINNs type regularization loss). Whereas in the linear case, the function is explicitly constrained to the space where the symmetries are satisfied, which is a stronger inductive bias.

---

> > > ### Author Response · Authors · 2024-08-12
> > >
> > > Thank you for your thoughtful response! We agree that the regularization for non-linear symmetry does not *explicitly* constrain the parameter space. However, despite its difference from how we handle linear cases, this regularization approach is still shown to be effective in several dynamical systems in Sec 5.3 and Table 4 from the rebuttal PDF. By encouraging the learned equation to conform to the symmetry, even if not perfectly, we are much more likely to discover the correct equation. We will leave the problem of building explicit constraints for non-linear symmetries to future works.

---

### Official Review · Reviewer_wQw7 · 2024-07-12

**Soundness:** 3
**Presentation:** 3
**Contribution:** 3
**Rating:** 7
**Confidence:** 3

**Summary:**

The paper proposes to leverage symmetry to guide the equation discovery process, compress the equation search space, and improve the accuracy and simplicity of the learned equations. Depending on the types of symmetries, the paper develops a pipeline for incorporating symmetry constraints into various equation discovery algorithms, including sparse regression and genetic programming

**Strengths:**

1. The paper clearly establishes a pipeline to use Lie point symmetries of ODEs to constrain the equation learning problem.

2. The paper theoretically derives the criterion for symmetry of time-independent ODEs in terms of equivariance of the associated flow map.

3. From the above mentioned criterion, the paper solves the constraint explicitly to compress the equation search space in sparse regression and promote parsimony in the learned equations.

4. In experiments across many dynamical systems with substantial noise, the symmetry-informed approach achieves higher success rates in recovering the governing equations.

**Weaknesses:**

The Time-Reversal Symmetric Ordinary Differential Equation Network has been proposed [1]. However, this work does not cite the relevant article [1]. While the presentation may differ, the symmetries investigated in both studies are fundamentally the same. Although this work is based on SINDy and  the relevant article [1] is based on Neural ODE, this may not be an essential difference. Therefore, the reviewer recommends a detailed comparison of the two approaches, highlighting their similarities and differences. Additionally, incorporating comparisons into the experimental section would be beneficial.

[1] Huh, I., Yang, E., Hwang, S. J., & Shin, J. (2020). Time-reversal symmetric ode network. Advances in Neural Information Processing Systems, 33, 19016-19027.

**Questions:**

1. Why set 5% and 20% noise respectively in the experiment in Section 5.1? What happens when the error scale is smaller?

2. For equation discovery with unknown symmetry, if we assume that the symmetry is unknown in the first two examples, what will be the result?

3. The paper only considers two-dimensional situations. What are the results for high-dimensional examples?

**Limitations:**

Yes

---

> ### Author Rebuttal · Authors · 2024-08-06
>
> We thank the reviewer for the valuable feedback and the recognition of our paper’s presentation and evaluation. We address the key comments below.
>
> > Related work: Time-reversal symmetric ODE Network (TRS-ODEN)
>
> Thank you for bringing this to our attention. This work aims to learn the physical dynamics with black-box neural networks whereas our goal is to discover the *symbolic* form of the governing equation. Furthermore, the range of symmetries considered in these two works differs. We consider the Lie symmetries of the ODE systems, which cover the broad range of continuous symmetry transformations.TRS-ODEN focuses on the specific symmetry of time reversal, which is a discrete symmetry. Because of the distinct natures of these symmetries, the ways of incorporating them also differ.
>
> For long-term prediction tasks, TRS-ODEN can be used as a baseline. In **Figure 5** (supplementary pdf), we tested TRS-ODEN and other NeuralODE-based methods on two tasks in our paper. In the damped oscillator, because it is irreversible and not conservative, TRS-ODEN and HODEN do not help. In the reversible Hamiltonian system of Lotka-Volterra equations, TRS-ODEN and HODEN improve upon the baseline Neural ODE, but have slightly higher errors than our discovered equations with Lie symmetry regularization.
>
> We agree that our work and TRS-ODEN cover different important aspects of ODE symmetries for different tasks. We will cite this work and explain the connections in the revised paper.
>
> > Effect of noise levels in Sec 5.1
>
> **Figure 4** in the supplementary pdf shows the equation discovery statistics under smaller noise levels. It is observed that the baseline without symmetry deteriorates quickly as the noise increases, while our symmetry-based methods constantly discover the correct equations.
>
> > If we don’t know the symmetry in the first two examples, what will be the result?
>
> In this case, we need to discover the symmetry first. We use LieGAN [1] (without the autoencoder) to learn the linear symmetry in these examples. For the damped oscillator, we learn the symmetry $v=0.97x_2\partial_1-x_1\partial_2$. The resulting equivariant basis is close to the one corresponding to the true rotation generator. Solving this symmetry constraint gives similar equation discovery results (**Figure 4** left, red line). For the growth model, the learned symmetry is $v=1.95x_1\partial_1+x_2\partial_2$. Solving the constraint in eq (10) also results in 3 close-to-zero singular values and we obtain exactly the same equivariant basis as in Figure 2 (lower) in the paper. Therefore, the equation discovery results are also the same.
>
> [1] Generative Adversarial Symmetry Discovery. ICML 2023.
>
> > High-dimensional examples
>
> In the supplementary PDF (**Table 4**), we provide two examples of learning higher-dimensional dynamics by solving the hard constraint (SEIR) and symmetry regularization (Lorenz).
>
> The SEIR equations [1] model the evolution of pandemics by the number of susceptible, exposed, infected, and recovered individuals. It is a 4-dimensional ODE system with quadratic terms. We consider the following equations:
>
> $
> \\left\\{
> \begin{aligned}
> \dot S &= 0.15-0.6SI\\\\
> \dot E &= 0.6SI-E\\\\
> \dot I &= E-0.5I\\\\
> \dot R &= -0.15+0.5I
> \end{aligned}
> \\right.
> $
>
> We assume an intuitive symmetry of scaling the number of recovered individuals proportional to the total population: $v=(S+E+I+R)\partial_R$. We solve the constraint w.r.t this symmetry as in Sec 4.1. **Figure 2** (supplementary PDF) shows the reduced parameter space (from 60D to 34D).
>
> For the 3D Lorenz system [2], we use parameters $\sigma=0.5,\beta=1.0,\rho=0.0$. Similar to Section 5.3, we discover the symmetry first and use the learned symmetry to regularize equation discovery.
>
> Please let us know if you have additional questions. If your concerns have been addressed, may we kindly request you increase the score for our paper?

---

> > ### Comment · Reviewer_wQw7 · 2024-08-10
> >
> > Thank you for your response. I believe the authors responded well to my questions, and this paper is a valuable contribution to the area of learning dynamical systems. I have improved my score.

---

### Official Review · Reviewer_jBog · 2024-07-12

**Soundness:** 3
**Presentation:** 3
**Contribution:** 3
**Rating:** 6
**Confidence:** 3

**Summary:**

This paper focuses on incorporating symmetries to equation discovery pipelines for ODEs.

If the governing equation has a known linear symmetry (its solutions are invariant with respect to a known linear action by a known Lie group), this paper derives a set of conditions the ODE needs to satisfy, and incorporates those conditions into a symbolic regression framework such as SINDy.

When the symmetry is still known but not linear, the paper derives a form of regularization to promote that symmetry into the learning process. This procedure wasn't completely clear to me.

When the symmetry is not known, the paper uses LaLiGAN, a symmetry discovery framework, to learn the corresponding regularization.

**Strengths:**

This is a well-written paper in an interesting topic: imposing symmetries in governing equation discovery.

It considers approaches from the simpler case (known linear symmetries) to the more general case (unknown non-linear symmetries), and it provides numerical evaluations for them.

**Weaknesses:**

The methodology behind the regularization approach and the LaLiGAN approach were not clearly explained in the paper. It would be good if the authors can provide a more detailed explanation.

In particular the technical approach for unknown non-linear symmetries (based on LaLiGAN) was not clear to me. I went to the recent LaLiGAN paper to see how it works and I found that the original LaLiGAN paper already proposes the approach for governing equation discovery. I suggest the authors explain how the approach they propose in this paper differs from the approach proposed in LaLiGAN. Moreover, the LaLiGAN paper does not have a github repository with their code, so the authors should be able to make sure that all their code and their dependencies can be made publicly available if the paper gets accepted.

**Questions:**

What are the limitations of the approach in the known non-linear symmetries case?

What are the limitation of the approach in the unknown symmetries case?

**Limitations:**

The authors could comment on the technical limitations of their work (e.g. dimensionality, ODE order, sample complexity, etc).

---

> ### Author Rebuttal · Authors · 2024-07-31
>
> We thank the reviewer for the valuable feedback. We address key comments below.
>
> > On the connection and difference from LaLiGAN
>
> **Our approach has a completely different goal from LaLiGAN.** Our method aims to discover equations using symmetry as an inductive bias. LaLiGAN aims to discover unknown symmetry.
>
> Connection: Our method requires the knowledge of symmetry. We consider two scenarios: one where symmetry is given to us as a prior, the other where we do not know the symmetry. Only in the latter case, **we use LaLiGAN as a tool to discover the symmetry of dynamical systems.** Our method can also work with any other symmetry discovery techniques.
>
> In terms of equation discovery, we propose equivariant models for this task given the symmetry, while LaLiGAN did not discuss equivariant models or any other new method for equation discovery.
>
> To explain the differences more specifically:
> * In Sec 5.1, we solve the equivariance constraints for linear symmetries, such as rotation and scaling. This is one of our new contributions and has not been discussed in LaLiGAN.
> * In Sec 5.2, we use LaLiGAN to discover a latent space (symmetry discovery) and then discover an equivariant equation in the latent space. In the second step (equation discovery), we enforce the equivariance constraint on the equation, whereas LaLiGAN uses a non-equivariant model to discover the equation. In other words, if we view the latent dynamics as the data for equation discovery, LaLiGAN has **equivariant data + non-equivariant model**, while we have **equivariant data + equivariant model**. In Figure 3, we show that using such an equivariant model that matches the symmetry of the data could further improve accuracy.
> * In Sec 5.3, we use LaLiGAN to discover nonlinear symmetry (same as LaLiGAN). Then, we encourage our model to be (approximately) equivariant to this symmetry through regularization (Sec 4.2). LaLiGAN did not use an equivariant model to discover equations. Same as in the above bullet point, they used equivariant data (in the latent space) + non-equivariant model. Moreover, they only learned equations in terms of latent variables. Our method discovers equations in the original state space, which is more interpretable.
>
> We will clarify these differences in the updated version of the paper.
>
> > Code availability
>
> In the Supplementary Material, we have included our implementation of LaLiGAN. Our code is fully runnable without external non-public dependencies.
>
> > Limitations in the case of known nonlinear symmetry / unknown symmetry
>
> When a nonlinear symmetry is known, we can directly apply the symmetry regularization approach, as shown in Sec 4.2. This would be a relatively easy scenario compared to our experiments in Sec 5.3, because we don’t need to discover the symmetry. There might be limitations to the regularization approach itself. For example, choosing the regularization coefficient may be cumbersome; the optimization problem may become more difficult due to non-convexity.
>
> When the symmetry is unknown, we use symmetry discovery methods such as LaLiGAN to learn the symmetry. If the discovered symmetry is inaccurate, it would cause an additional source of error in computing our symmetry regularization. Thus, our method indeed relies on the accuracy of the learned symmetry.
>
> > Other technical limitations
>
> We thank the reviewer for pointing out these potential limitations. We provide some brief comments below and will discuss these points in more detail in the revised paper.
>
> Regarding dimensionality, our current experiments mainly focus on low-dimensional problems. Symmetries in higher-dimensional systems can be more challenging to identify. However, once we know the symmetry, our method can be easily extended to higher-dimensional problems. Also, higher-order systems can be reduced to lower order by introducing the higher-order derivatives as new state variables.

---

> > ### Comment · Reviewer_jBog · 2024-08-12
> >
> > Thank you for the response. I keep my positive score of the paper.

---

### Author Rebuttal · Authors · 2024-08-06

We thank the reviewers for their detailed and valuable feedback. We are encouraged that they find our work to be a clearly motivated idea (R2,3) towards the interesting problem of equation discovery (R1). We are also glad that they find our paper well-written and easy to follow (R1,4), our method principled and theoretically grounded (R2,4), and demonstrated to be effective in a wide range of tasks (R2,3).

## Common questions

Here we answer some common questions from the reviewers. This is a high-level summary; more detail can be found in individual responses.

> On the connection and difference from LaLiGAN

Our work has a completely **different goal** from LaLiGAN. Our method aims to discover equations using symmetry as an inductive bias. LaLiGAN aims to discover unknown symmetry. Only when we do not know the symmetry a priori, we use LaLiGAN as a tool to discover the symmetry first and use the discovered symmetry to regularize equation learning.

Also, our work has made important adaptations to incorporate learned symmetry into equation discovery, e.g. extracting the infinitesimal action in eq (12) and using relative regularization loss in eq (11).

> The types of symmetries in different experiments

There is some confusion about what kind of symmetry we are using for each experiment. In particular, in Sec 5.3, we use *nonlinear* symmetries *discovered* by LaLiGAN for symmetry regularization. It is *not* true that the symmetry is linear (R2), or there is no symmetry in this case (R5).

We believe the above information is available in the paper. However, we will provide more clarifications in the revision to further avoid confusion.

## The supplementary PDF

Here we also provide an index of the supplementary PDF. It includes additional illustrations and tables of experiment results requested by the reviewers.
* Table 1: an ablation study for the relative regularization on the Lotka-Volterra equations. It is shown that the relative loss leads to a smaller parameter estimation error than the absolute loss.
* Table 2: refining symmetry-regularized solutions. The *learned* symmetries may introduce slight biases. This problem can be addressed by removing the symmetry regularization, fixing the equation form, and optimizing the equation parameters under the equation $L_2$ loss only. This process reduces the parameter estimation error.
* Table 3: using combined regularization terms can sometimes lead to better discovery. This is motivated by the idea of Sobolev training, thanks to the reference from R4.
* Table 4: equation discovery on high-dimensional systems, as requested by R2 and R4.
* Figure 1: an intuitive demonstration of how symmetry helps the search in the equation parameter space. The equation parameter space is abstracted into a 2D plane, where the symmetry constraint leads to a smaller 1D subspace and makes optimization easier.
* Figure 2: the equivariant basis for the SEIR model in Table 4. In this case, symmetry reduces the 60D parameter space to 34D, significantly reducing the complexity of the optimization problem.
* Figure 3: experiments in Sec 5.1 with different noise levels.
* Figure 4: Comparison with Neural-ODE-based methods, including TRS-ODEN, as requested by R2.

---

### Decision · Program_Chairs · 2024-09-25

**Decision:**

Accept (poster)

**Comment:**

This paper enhances equation discovery by incorporating symmetry constraints from Lie groups, improving accuracy and simplicity. The method, tested on various dynamical systems, shows superior performance over baseline approaches by leveraging symmetry to refine and constrain the search space. Some confusion about the difference to LaLiGAN (which also proposes a simpler equation discovery method) can and should be clarified in the rebuttal and should be made very clear in the camera ready version. The rebuttals have resolved other reviewer concerns, and the proposed changes are straightforward for the final version. I concur with the reviewers' consensus on acceptance. Ensure the camera-ready version also clearly and early explains that the linear case restricts the solution space while the non-linear case introduces an inductive bias via symmetry regularization, as discussed with Reviewer zT4E.